evolution, theoretical biology, behaviour

life-history theory, ageing, maternal allocation, energy dynamics, stochastic dynamic programming

**Author for correspondence:**
Antoine M. G. Barreaux
e-mail: antoine.barreaux@gmail.com

Electronic material is available online at https://doi.org/10.6084/m9.figshare.c.5820931.

# Incorporating effects of age on energy dynamics predicts nonlinear maternal allocation patterns in iteroparous animals

Antoine M. G. Barreaux[1,2,3], Andrew D. Higginson[4], Michael B. Bonsall[5,6] and Sinead English[1]

[1]School of Biological sciences, University of Bristol, Bristol BS8 1TQ, UK
[2]CIRAD, UMR INTERTRYP, F-34398 Montpellier, France
[3]INTERTRYP, Univ Montpellier, CIRAD, IRD, 34000 Montpellier, France
[4]Centre for Research in Animal Behaviour, College of Life and Environmental Sciences, University of Exeter, Exeter EX4 4QG, UK
[5]Department of Zoology, Mathematical Ecology Research Group, University of Oxford, Oxford OX1 3PS, UK
[6]St Peters College, Oxford OX1 2DL, UK

AMGB, 0000-0001-5822-761X; ADH, 0000-0002-2530-0793; MBB, 0000-0003-0250-0423; SE, 0000-0003-2898-2301

Iteroparous parents face a trade-off between allocating current resources to reproduction versus maximizing survival to produce further offspring. Parental allocation varies across age and follows a hump-shaped pattern across diverse taxa, including mammals, birds and invertebrates. This nonlinear allocation pattern lacks a general theoretical explanation, potentially because most studies focus on offspring number rather than quality and do not incorporate uncertainty or age-dependence in energy intake or costs. Here, we develop a life-history model of maternal allocation in iteroparous animals. We identify the optimal allocation strategy in response to stochasticity when energetic costs, feeding success, energy intake and environmentally driven mortality risk are age-dependent. As a case study, we use tsetse, a viviparous insect that produces one offspring per reproductive attempt and relies on an uncertain food supply of vertebrate blood. Diverse scenarios generate a hump-shaped allocation when energetic costs and energy intake increase with age and also when energy intake decreases and energetic costs increase or decrease. Feeding success and environmentally driven mortality risk have little influence on age-dependence in allocation. We conclude that ubiquitous evidence for age-dependence in these influential traits can explain the prevalence of nonlinear maternal allocation across diverse taxonomic groups.

## 1. Introduction

Maternal allocation of resources to offspring typically has a positive effect on offspring traits such as longevity and fecundity and thereby offspring fitness [1–4]. However, as resources are limited, allocation can negatively affect maternal survival and future reproduction [5–8]. Mothers therefore face a trade-off between current and future reproductive allocation. Maternal physiological state influences this allocation trade-off, and maternal state can vary with age, as foraging efficiency may decrease [9], physiological functions decline and cellular damage accumulates [10–13], with the result that mothers can face an increased risk of death as they get older [1,14].

In many systems, maternal allocation tends to decline with age, termed reproductive senescence [10,15,16]. Explanations of such senescence are based on the declining strength of natural selection with age [6], permitting the accumulation of deleterious mutations [17] or the fixation of alleles with a pleiotropic effect that favour fitness early in life but have negative effects later on [18]. A particular formulation of such explanations is the 'disposable soma',

whereby investment in early-life fitness traits is traded off against maintenance and later-life survival and reproduction [8]. High allocation of resources into early reproduction may be favoured under high mortality risk, but then damage accumulation and increasing constraints on resource acquisition reduce reproduction as an individual ages [19]. Alternatively, reproductive restraint in later life may be an adaptive strategy to cope with the accumulation of reproductive damage and the associated increase in mortality [20].

Allocation of resources by iteroparous females often increases and then decreases with age, a nonlinear allocation pattern observed across taxonomic groups (mammals, birds, invertebrates) in both laboratory and wild populations [4,10,15,21–23]. For younger mothers, allocation may increase with age as mothers gain experience in breeding and in acquiring food [7,15,22,24]. Allocation may then subsequently decline in later life due to the drivers mentioned above. To our knowledge, only one study predicted an increase and decrease in fecundity as a by-product of natural selection [25] and few—if any—theoretical studies predict this lifetime reproductive resources allocation pattern, from an initial increase to a later-life decline.

Most iteroparous females face some uncertainty in energy dynamics in terms of acquiring resources and using them (e.g. metabolic costs) [26,27]. Energy dynamics can also change with age due to the effects of experience [7,15,22], damage accumulation [10–13] and declining movement ability [9]. Most models do not incorporate how variation in energy dynamics [15,19] impacts the evolutionary strategy of resource allocation, although stochasticity has been shown to impact life histories, with different phenotypes being optimal in stochastic versus constant environments [28]. Models that consider variation in resource acquisition do not incorporate stochasticity or age-dependence in food availability and acquisition costs [29]. To our knowledge, only two models have shown how environmental uncertainty influences optimal maternal allocation under scenarios of varying food availability [26,27], modifying the balance between reproduction, maintenance and energy storage. These studies did not, however, link stochasticity to age-dependent allocation.

Current models may also be inadequate as they tend to focus on offspring number rather than offspring quality [19,20,30–32], which may underestimate the extent of reproductive senescence. The focus on offspring number is, in part, due to classic evolutionary theory of ageing assuming that all offspring are of equal quality (in terms of their lifetime reproductive success) [6,8]. However, offspring quality can decrease with maternal age because of epigenetic factors, changing offspring environment or constraints on maternal resource acquisition and allocation [22,33]. Later-born offspring have lower fitness and less relative reproductive value to parents, which increases the steepness of the age-related decline in the strength of natural selection and hence of reproductive senescence [32,34,35].

Here, we investigate how stochasticity and age-dependence in energy dynamics influence maternal allocation in iteroparous females. We develop a state-dependent model to calculate the optimal maternal allocation strategy with respect to maternal age and energy reserves, focusing on allocation in a single offspring at a time. We introduce stochasticity in energetic costs—in terms of the amount of energy required to forage successfully and individual differences in metabolism—and in feeding success. We systematically assess how allocation is influenced by age-dependence in energetic costs, feeding success, energy intake per successful feeding attempt and environmentally driven mortality.

We use, as a case study, a viviparous and iteroparous insect, tsetse (Glossina spp.). Tsetse are relatively long-lived flies, surviving up to three months in the wild [36–38], with the potential for allocation trade-offs between each birth similar to long-lived vertebrates [10,15,16]. Reproduction is highly costly for tsetse as mothers can give birth to offspring as large as themselves [36,39]. Maternal allocation is key for offspring survival [40,41], as there is no self-feeding after birth: larvae pupate, relying on maternal reserves until emerging as an adult 20–30 days later [38,42]. Tsetse have access to a rich food supply—vertebrate blood [42]—but this can be highly uncertain, requiring finding a host and avoiding its defences (e.g. swatting) [36,43], which introduces stochasticity in the flight duration and distance as well as the bloodmeal volume and hence in the energetic costs of flight and blood digestion [36,38,43]. While there was previously mixed support for age-dependent maternal allocation in tsetse [44–46], a recent laboratory study demonstrated hump-shaped allocation with age in Glossina morsitans morsitans [23].

Using our model, we identify scenarios with an optimal resource allocation strategy that leads to a hump-shaped maternal allocation in iteroparous females. We show that this nonlinear allocation pattern emerges in diverse scenarios, and the wide-ranging empirical evidence for age effects on the traits involved can explain why nonlinear allocation is found across numerous iteroparous animals.

## 2. Methods

### (a) The model
Using stochastic dynamic programming [47,48], we calculate the optimal amount of reserves $M$ that mothers allocate to each offspring depending on their own reserves $R$ and age $A$. The optimal life-history strategy is then the set of allocation decisions $M(R, A)$ over the whole lifespan that maximizes the total reproductive success of distant descendants. All model parameters and values are described in table 1. The model set-up allows the optimal strategy to be anywhere on the continuum from extreme capital (females build up stored reserves that are used for reproduction) to extreme income breeding (females do not store reserves across breeding events and use only those acquired during feeding), given their ecology [49].

### (b) State variables
Maternal reserves $R$ take values between $R_{min}$ (at or below which an individual dies) and $R_{max}$. Here, units of reserves are arbitrary but as an example could represent milligrams of fat, as this is the major macronutrient allocated from mother to offspring [50]. Each mother's lifespan is divided into time periods ($t$), equivalent to the time needed to complete one reproductive cycle. A mother's age $A$ is thus the number of time periods $t$ multiplied by the duration of each period. The model was parameterized such that these limits ($R_{max}$, $A_{max}$) do not influence the strategy because they are unlikely to be reached.

**Table 1.** Optimal allocation strategy model parameters. Parameter values for tsetse in the baseline and range of values explored during model evaluation ('—' means no range of values were explored for that parameter).

| symbol | description | baseline value | range explored |
|---|---|---|---|
| **variables** | | | |
| $R$ | reserve state | 0 to 50 | — |
| $A$ | age with $A_t = 9 \times t$ | 0 to 270 | — |
| $M$ | maternal allocation decision | 0 to 50 | — |
| **parameters** | | | |
| $t$ | time period (number of reproductive cycles) | 1 to 30 | — |
| $T_{max}$ | maximum time period | 30 | — |
| $R_{min}$ | reserve level at or below which individuals die | 0 | — |
| $R_{max}$ | maximum level of reserves | 50 | — |
| $d$ | environmentally driven mortality rate | 1/11 | 0.09 to 0.21 |
| $q$ | probability of successfully feeding $\{t = 1, t > 1\}$ | $\{0.35; 0.9\}$ | $\{0.35; 0.54$ to $1\}$ |
| $z$ | number of feeding opportunities per time period | 4 | — |
| $y$ | energy gained per successful feeding attempt | 6 | 2 to 13 |
| $c$ | energetic costs | 7 | 1 to 42 |
| $p$ | energy required to survive the non-feeding phase | 8 | — |

## (c) Trait dynamics

Reserves $R$ vary linearly over time, and at the start of period $t + 1$ $R$ is:

$$R_{t+1_{n,j}} = R_{t_{n,j}} - M_t + ny - (c + j) \quad \text{and}$$
$$n = [0 \dots z] \text{ and } j = \{-1, 0, 1\}, \tag{2.1}$$

$R_t$ denotes the maternal reserves at the start of period $t$, $M_t$ is maternal allocation for period $t$, $ny$ is the total energy intake ($y$ units of energy per successful feeding attempt, $n$ times per period), $c$ denotes energetic costs of basal metabolism, food-seeking movements and egg production and $j$ represents stochasticity in costs (costs being $c - 1$, $c$ or $c + 1$ with probability 0.25, 0.5 or 0.25, respectively). An individual has $z$ feeding opportunities per time period with a probability $q_t$ of success (which may vary with age). Feeding opportunities are assumed to be independent, so the probability of successfully feeding $n$ times is:

$$P(n) = q_t^n \times (1 - q_t)^{z-n} \quad n = [0 \dots z]. \tag{2.2}$$

## (d) Offspring production

If the optimal decision is not to allocate any resources ($M_t^* = 0$), then no offspring can be produced. If resources are allocated ($M_t^* > 0$), then a juvenile offspring is produced. Here, we consider a specific case when offspring rely on maternal reserves for survival until maturity. As such, if maternal allocation $M_t^*$ exceeds the energy required to survive the non-feeding phase until adulthood ($p$), an adult offspring is produced. The reserves at the start of adulthood $R_1$ of this offspring are equal to the maternal allocation $M_t^*$ minus $p$.

## (e) Mortality

Mothers die when they run out of reserves ($R_t \leq R_{min}$). They also face environmentally driven mortality at rate $d$, for example, the risk of dying from predation or inclement weather.

## (f) Fitness calculation

The decisions are found working backwards from $t = T_{max}$ (assuming zero fitness at $T_{max}$). The expected fitness $h$ of a mother at time period $t$, given her reserves $R$, age $A$ and allocation $M$ is:

$$h(R_t, A_t \mid M_t) = f(M_t - p) + (1 - d_t)$$
$$\cdot \sum_{n=0}^{z} \sum_{j=-1}^{1} (w(R_{t+1_{n,j}}) \cdot v(R_{t+1_{n,j}}, A_{t+1}) \cdot q_{n,j} \cdot c_{n,j}). \tag{2.3}$$

Fitness is the sum of the immediate gain in fitness $f$ (of producing an adult offspring) and the expected future reproductive success $v$. The expected future reproductive success $v$ is conditional on individuals avoiding environmentally driven mortality, $(1 - d_t)$, and not starving to death, $w > 0$, which depends on the state of maternal reserves at the start of the next time step $R_{t+1_{n,j}}$. The future state $R_{t+1_{n,j}}$ depends on the decision $M_t$, as well as the probabilistic feeding success $q_{n,j}$ and costs $c_{n,j}$.

$$w(R_{t+1_{n,j}}) = \begin{cases} 0, & R_{t+1_{n,j}} \leq R_{min} \\ 1, & R_{t+1_{n,j}} > R_{min} \end{cases}. \tag{2.4}$$

At each time, the expected future reproductive success $v$ is obtained from the allocation $M_t$, which maximizes $h$ for a given state of maternal reserves $R$ and age $A$:

$$v(R_t, A_t) = \max_M \{h(R_t, A_t \mid M_t)\}. \tag{2.5}$$

More information about the optimization process can be found in the electronic supplementary material.

The immediate gain in fitness $f$ is a function of offspring quality (i.e. the expected reproductive success during the offspring's lifetime given its energy reserves at the start of adulthood) and depends on maternal allocation $M_t$ minus the energy needed to survive the non-feeding phase $p$. To calculate $f$, we run repeated backwards iterations (over several generations [51]), initially assuming a Gompertz function for $f$:

$$f(M_t) = k_1 e^{-k_2 e^{-k_3(M_t - p)}} \text{ where } k_1 = 1, k_2 = 5, \text{ and}$$
$$k_3 = 0.15. \tag{2.6}$$

We obtain an array containing the values of fitness $v$ for a given state, $v(R, A)$. Keeping only values at the first time period

**Table 2.** Age-dependent parameter variation. Linear or asymptotic age-dependent functions of energetic costs ($c_t$), probability of successfully feeding ($q_t$), energy gained per successful feeding attempt ($y_t$) and environmentally driven mortality ($d_t$).

| age dependence | equation | values | rationale |
|---|---|---|---|
| **energetic costs $c_t$** | | | |
| increasing linearly | $c_t = c_1 + c_2 \times t$ | $c_1 = 2, 4$ or $6$ | increasing difficulties in host searching and flying as |
| | | $c_2 = 0.5, 1, 1.5, 2, 2.5$ or $3$ | damage accumulates [11,12] |
| decreasing asymptotically | $c_t = c_1 \times (1.2 + e^{-c_2 \times t})$ | $c_1 = 1, 2, 3, 4, 5$ or $6$ | no development costs of flight muscles, thoracic cuticle, or reproductive structures post maturity [58,59,62]; |
| | | $c_2 = 0.5$ or $1$ | increased vision at maturity [63] improves host searching |
| **energy gained per successful feeding attempt $y_t$** | | | |
| increasing linearly | $y_t = y_1 + y_2 \times t$ | $y_1 = 6$ | energy transfer efficiency increases past first reproduction |
| | | $y_2 = 0.1, 0.2, 0.3, 0.4$ or $0.5$ | or digestion improves |
| increasing asymptotically | $y_t = y_1 \times (1 - e^{-y_2 \times t})$ | $y_1 = 6, 7, 8, 9, 10$ or $11$ | fully developed gut at maturity with more volume for |
| | | $y_2 = 0.5, 1, 1.5$ or $2$ | blood [39] |
| decreasing linearly | $y_t = y_1 - y_2 \times t$ | $y_1 = 6, 7, 8, 9, 10$ or $11$ | digestion decreases because of gut deterioration |
| | | $y_2 = 0.1, 0.2$ or $0.3$ | |
| **probability of successfully feeding $q_t$** | | | |
| increasing asymptotically | $q_t = q_1 / (0.9 + e^{-q_2 \times t})$ | $q_1 = \{0.35; 0.9\}$ for $\{t = 1, t > 1\}$ | experience increasing host searching and host defence |
| | | $q_2 = 0.5, 1, 1.5$ or $2$ | escape |
| decreasing linearly | $q_t = q_1 - q_2 \times t$ | $q_1 = \{0.35; 0.9\}$ for $\{t = 1, t > 1\}$ | host searching decreases as olfaction decreases with age |
| | | $q_2 = 0.01, 0.02$ or $0.03$ | [64] |
| **environmentally driven mortality rate $d_t$** | | | |
| increasing linearly | $d_t = d_1 + d_2 \times t$ | $d_1 = 1/11$ | flying ability decreases as damage accumulates [11,12], |
| | | $d_2 = 0.002, 0.004, 0.006, 0.008$ or $0.010$ | increasing predation and host swatting risks |

$t = 1$, we obtain the fitness of offspring upon reaching adulthood. We divide this value by the highest fitness value (in this iteration at $t = 1$) to keep the fitness between 0 and 1. We run the backward iteration for the next generation to calculate the updated fitness of offspring produced upon reaching adulthood, $f$:

$$f(M_t - p) = v(R_1, A_1) \, / \, \max\{v(R_1, A_1)\} \text{ for all } R_1 \text{ and with } R_1$$
$$= M_t - p. \tag{2.7}$$

We repeat that process for 10 generations, by which time the frequency distribution of the $f$ values converges (electronic supplementary material).

## (g) Forward simulation

We simulated the life histories of 1000 mothers (electronic supplementary material, code modified from [52]) following the optimization strategy and the reserves at the start of adulthood $R_1$, the distribution of which was determined using an iterative procedure as described in [53] (electronic supplementary material). For each individual, we calculated maternal allocation $M_t$, maternal reserves $R_t$ and relative allocation $M_t/R_t$ at each time period $t$ to understand how resources are partitioned between mother and offspring.

## (h) Model assumptions for tsetse

Each reproductive cycle ($t$) is nine days long, from egg laying *in utero* to birth, as observed in *G. morsitans morsitans* at 25°C [54,55]. We set $M_t$ to zero for the first two time periods, as it

takes 18–20 days before the first offspring is produced in the wild [56] and a mother gives birth to her first offspring at the start of the third time period (19–20 days, see details in electronic supplementary material). The maximum lifespan is set at $A_{\max} = 270$ days as, in the wild, individuals live on average 60–90 days, and fewer than 1% survive beyond 270 days [37,38,42]. Environmentally driven mortality is set such that two-thirds of individuals are expected to die before reaching 100 days old, as in the wild, mortality reaches 90% by 100 days [37,38,42]. Tsetse have four feeding opportunities per time period $t$ (every 2–3 days [42]). The feeding success $q_t$ is lower when $t = 1$ ($q_1 = 0.35$), as newly emerged tsetse are relatively inactive up to 2 days post emergence [57], and flight muscles take 8–10 days to fully develop [58–60]. After this point, feeding success is high, $q_t = 0.9$ (see details in electronic supplementary material), given stronger host detection abilities [61].

## (i) Model evaluation

We consider how the optimal strategy varies when there is age-dependence in resource acquisition, energetic costs and survival. Specifically, we include varying scenarios with an age-dependent increase or decrease in energetic costs ($c_t$), feeding success ($q_t$), energy intake per successful feeding attempt ($y_t$) and environmentally driven mortality rate ($d_t$) (table 2). We consider the age-dependence of parameters one at a time or in pairs (table 3), altering the slope, intercept or asymptote of the age-dependent function (linear or asymptotic function). The parameter space exploration was designed so that no age-dependent parameter would cause the net gain in resources to fall below $R_{\min}$ or above $R_{\max}$ before 100 days of age, and there

**Table 3.** Model evaluation. Scenarios with age-dependent parameters, individually or in pairs, and with a quadratic downward model being the better fit to the simulated maternal allocation data (proportions in brackets). The goodness-of-fit is also provided with the pseudo $R^2$ conditional value (proportion of variance explained by the fixed and random terms for the model fit, accounting for individual identity) being above 0.7 or not (proportions in brackets). The parameters varying are the energetic costs ($c_t$), probability of successfully feeding ($q_t$), energy gained per successful feeding attempt ($y_t$), and environmentally driven mortality ($d_t$) (table 2).

| age-dependent parameters | | better fit quadratic downward/ number of scenarios evaluated | conditional pseudo $R^2$ value above 0.7/number of scenarios evaluated |
|---|---|---|---|
| $c_t$ linear increase | | 7/18 (0.39) | 0/18 (0) |
| $c_t$ asymptotic decrease | | 1/12 (0.08) | 0/12 (0) |
| $y_t$ linear increase | | 0/5 (0) | 0/5 (0) |
| $y_t$ linear decrease | | 18/18 (1) | 1/18 (0.06) |
| $y_t$ asymptotic increase | | 10/24 (0.42) | 0/24 (0) |
| $y_t$ linear decrease | | 0/3 (0) | 0/3 (0) |
| $q_t$ asymptotic increase | | 0/4 (0) | 0/4 (0) |
| $d_t$ linear increase | | 0/5 (0) | 0/5 (0) |
| $c_t$ linear increase | $y_t$ linear increase | 39/90 (0.9) | 0/90 (0) |
| | $y_t$ linear decrease | 144/324 (0.44) | 14/324 (0.04) |
| | $y_t$ asymptotic increase | 297/432 (0.69) | 17/432 (0.05) |
| | $q_t$ asymptotic increase | 36/72 (0.5) | 1/72 (0.01) |
| | $q_t$ linear decrease | 12/54 (0.22) | 0/54 (0) |
| | $d_t$ linear increase | 39/90 (0.43) | 0/90 (0) |
| $c_t$ asymptotic decrease | $y_t$ linear decrease | 135/216 (0.625) | 2/216 (0.01) |
| | $q_t$ linear decrease | 11/36 (0.31) | 0/36 (0) |

was not 100% mortality before this time. Our aim is to identify whether the observed reproductive senescence can arise from optimal maternal allocation. As such, we do not impose a decline in selection in later life as all offspring are potentially equally valuable at all ages (for the same maternal allocation), and we assume there are no mutations. However, mothers may vary allocation of resources to offspring with age, which will then result in offspring of different quality and may lead to reproductive senescence if offspring quality decreases with maternal age.

For each scenario, we run the backward iteration process with these age-dependent functions, obtain the allocation strategy and simulate the life history of 1000 individuals based on the novel strategy. We then fit quadratic and linear models to the reproduction of these individuals using the lme function, nlme package [65] in R [66]. The response variable is the maternal allocation $M_t$, and explanatory variables are time period $t$ and $t^2$ (for the quadratic fit only), with individual identity as a random term.

We use likelihood ratio tests to compare linear and quadratic models using the *anova* function (package *nlme* [65]) with the maximum-likelihood method [67]. If the comparison is significant ($p$-value < 0.05), we considered the quadratic model to have a better fit, otherwise the linear model is considered more parsimonious. We were particularly interested in identifying scenarios where the fit was quadratic with a negative quadratic term, to understand nonlinear allocation patterns found in iteroparous animals in general [4,10,15,21,22] and tsetse in particular [23].

It is worth noting that caution is required when interpreting quadratic parameters in terms of senescence to infer an initial increase of reproductive performance until a peak or plateau followed by a decrease of reproductive performance. This is because the presence of a statistically significant negative quadratic coefficient does not necessarily indicate a hump-shaped curve but can also represent a case of diminishing returns where allocation plateaus in later life but does not decline (hence, no

reproductive senescence). In our case, we were confident that the negative quadratic term would be appropriate given that this was the best fit to our empirical data [23]—which were also analysed using more flexible approaches.

For each scenario, the pseudo $R^2$ conditional value (proportion of variance explained by the fixed and random terms, accounting for individual identity) is calculated to assess the goodness-of-fit of the lme model, on a scale from 0 to 1, using the 'r.squared' function, package gabtool [68,69].

## (j) Nonlinear maternal allocation in tsetse

To help explain the drivers behind the nonlinear allocation pattern observed in tsetse in the laboratory [23], we focused on model scenarios where the downward quadratic model fits the simulated allocation data with a pseudo $R^2$ conditional value above 0.7 (given the marked diminution in the frequency of scenarios with $R^2$ past that value; electronic supplementary material). We then selected the scenario where the fitted parameters for the quadratic fit were within a 90% CI of the parameters for the quadratic fit to the tsetse laboratory allocation data ($M_t = 10.41 + 5.20 \times t - 0.40 \times t^2$ [23]; electronic supplementary material).

## 3. Results

Exploring first the baseline case of the model, the optimal allocation decision is dependent on maternal reserves but independent of age (figure 1a, solid grey line). Individuals do not build up reserves across breeding events and instead allocate nearly all available reserves to each offspring, as the relative allocation is close to 1 (figure 1b,c, solid grey line). They maintain just enough reserves to make the risk of starvation negligible.

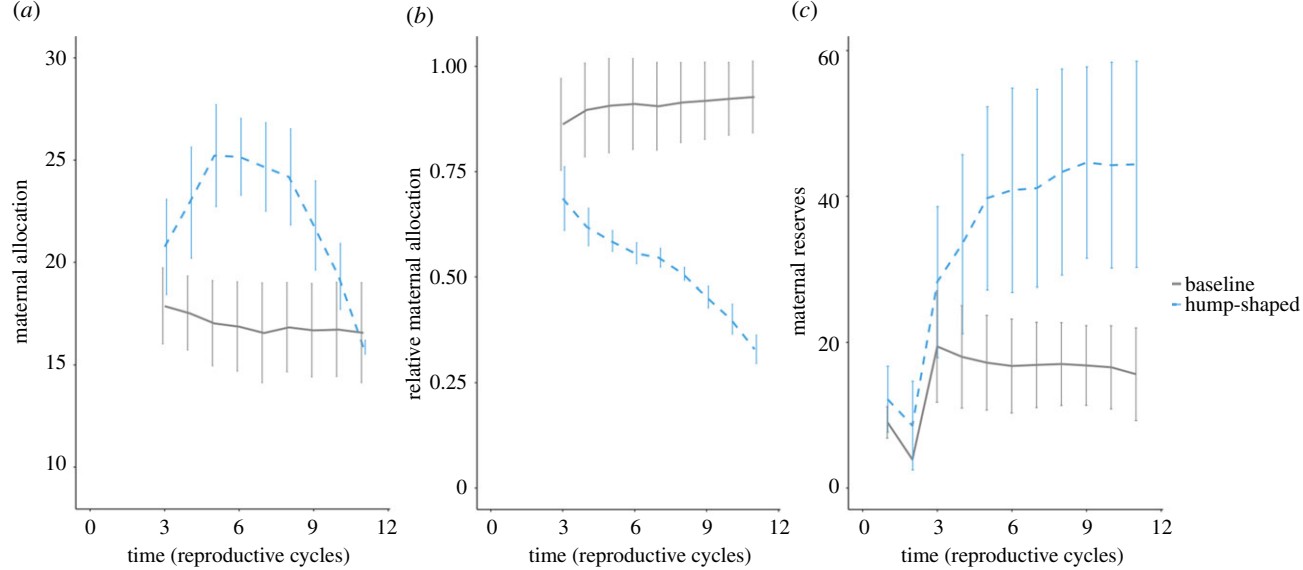

**Figure 1.** Maternal allocation (*a*), relative allocation (*b*) or maternal reserves (*c*) for the baseline model (solid grey line) or the selected tsetse hump-shaped allocation pattern (dashed sky-blue line). Average maternal or relative allocation or reserves of 1000 mothers for 12 reproductive cycles (x-axis). The error bars are the s.d. of the maternal or relative allocation or reserves, respectively. The relative allocation is the maternal allocation divided by maternal reserves. (Online version in colour.)

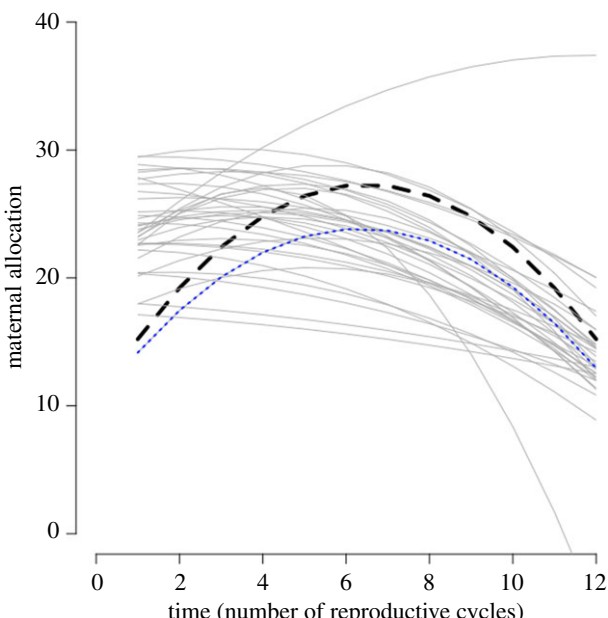

**Figure 2.** Scenarios with a good quadratic downward fit to the simulated allocation data. For 35 scenarios (solid grey lines), the quadratic downward model of the simulated allocation data was a better fit and had a conditional pseudo $R^2$ value above 0.7. The line in dotted blue depicts the scenario closest to the quadratic fit of the tsetse laboratory data (which is in dashed black) [23]. (Online version in colour.)

## (a) Model evaluation

We predicted a nonlinear (hump-shaped) pattern of allocation in 53% (749/1403) of the scenarios evaluated. Thirty-five of these quadratic downward scenarios fit the simulated data with a conditional pseudo $R^2$ value above 0.7 (table 3 and figure 2).

One scenario had a single parameter that is age-dependent: a linear decrease in energy intake per successful feeding attempt $y_t$. When considering age-dependence in two parameters, a hump-shaped allocation was primarily observed when energetic costs $c_t$ increased linearly in combination

with an asymptotic increase in energy intake $y_t$, a linear decrease in $y_t$ or an asymptotic increase in feeding success $q_t$. We also obtained a hump-shaped allocation with an asymptotic decrease in costs $c_t$, in combination with a linear decrease in energy intake $y_t$.

## (b) Nonlinear maternal allocation in tsetse

Considering age-dependence in parameters (tables 2 and 3), out of the scenarios for which a quadratic downwards model was the better fit and a conditional pseudo $R^2$ value above 0.7 of that fit, we selected one scenario (figure 2) that was the best fit to the laboratory data [23] (i.e. quadratic fit parameters within a 90% CI of parameters for the quadratic fit to the laboratory data [23]; electronic supplementary material), termed 'hump-shaped' (figure 1). In this scenario, energetic costs increase linearly ($c_t = c_1 + c_2 \times t$ with $c_1 = 2$ and $c_2 = 2$) and energy intake increases asymptotically ($y_t = y_1 \times (1 - e^{-y_2 \times t})$ with $y_1 = 11$ and $y_2 = 0.5$). The fitted function based on the simulations for the maternal allocation pattern is $M(t) = 10.19 + 4.31t - 0.34t^2$.

Maternal allocation decreases with age for a given level of reserves in the associated optimal allocation strategy (electronic supplementary material). Relative allocation decreases over time in the simulated allocation data (figure 1*b*, dashed sky-blue line), while reserves increase (figure 1*c*, dashed sky-blue line).

## 4. Discussion

Our model predicts that optimal maternal allocation of resources is nonlinear with age, when there is age-dependence in key drivers of energy dynamics. Such a nonlinear relationship between parental allocation and age has been found in many species. Our model assumes no mutations and hence provides further theoretical insight into the drivers of age-dependent allocation in terms of optimal life-history allocation, although we acknowledge that similar patterns can also arise from changes in mutation pressure that are not considered in

our model. The main parameters leading to a hump-shaped optimal allocation are a combination of age-dependence in energy intake and energetic costs. There is only one scenario where age-dependence in feeding success generated a hump-shaped allocation and no scenarios that included age-dependence in environmentally driven mortality. The scenario that best described the hump-shaped allocation observed in laboratory tsetse [23] included an asymptotic increase in energy intake combined with a linear increase in energetic costs.

In the context of theoretical models of maternal allocation and empirical evidence [4,7,15,19–23], our model confirms allocation to be age-independent without damage accumulation, age-dependence in key traits or a specific focus on the terminal investment near the end of a fixed lifespan [6,70–72]. Adding stochasticity into the energetic costs without age-dependence does not lead to age-dependent allocation. This fits with theoretical predictions that stochasticity in overhead costs of reproduction (for example reproductive structures like milk glands) is not sufficient to influence maternal allocation over time [73,74].

Our model confirms that a hump-shaped maternal allocation [4,10,15,21,23] can be an adaptive strategy in iteroparous animals, without specifically imposing a declining selection pressure with age, under a diverse set of scenarios. Specifically, optimal nonlinear allocation is found in scenarios with an increase in energy intake and energetic costs, and those with a decrease in energy intake on its own or combined with an increase or decrease in energetic costs. These scenarios confirm the impact on allocation of gains in experience in breeding and in acquiring food [7,15,22] and increasing energetic costs across the lifespan because of damage accumulation [7,15,19,22]. The evidence for age-dependence in such traits is wide-ranging across systems, from a decrease in energetic costs with an improved lactation ability (e.g. seals [24]), an improved energy transfer efficiency (rats [75]), or reduced metabolic requirements post maturation (tsetse [58,59,62]), to an increase in energy intake with an improved mobility post maturation (tsetse [58,59,63]) or a decrease in energy intake later in life because of gut deterioration (*Drosophila* [76]) or other physiological deteriorations. Such evidence could explain why these nonlinear patterns of maternal allocation are found across diverse taxonomic groups. Imposing a declining selection with age by relaxing the hypothesis that all offspring are equal may potentially nuance our predictions about nonlinear parental allocation. We hope our model will inspire future work on age-dependent allocation under varying assumptions about offspring quality.

Our model shows that age-dependence in feeding success is not a strong driver of a hump-shaped allocation, as only one such scenario had a good quadratic downward fit to the data. Previous studies have shown that small variations around intermediate levels of energy availability can lead to large non-monotonic changes in age-independent optimal allocation, but variation has less influence around low or high energy availability levels [26]. Age-dependence in feeding success in our model may drive energy availability to high or low levels, limiting variations in allocation and preventing hump-shaped patterns from being optimal strategies.

We found no effect of age-dependence in mortality on maternal allocation, in contrast to previous theoretical studies, where higher or lower age-independent mortality has been shown to shift the optimal allocation of resources from maintenance to reproduction [19]. This contrast could be explained by the fact that our model does not explicitly consider allocation towards maintenance *per se*, rather individuals maintain maternal reserves above $R_{min}$ to prevent condition-dependent death. We also do not consider damage accumulation depending on maintenance, which would increase the risk of damage-associated mortality with age, and potentially shift the allocation trade-off towards increased reproduction.

Our results represent the expected population-level average maternal allocation with respect to age, which may not necessarily capture the individual strategies. Stable individual differences in state-dependent adaptive behaviour have been shown to occur in another theoretical study in ecological contexts of intermediate favourability [77], which could be similar to what tsetse experience with rich food (vertebrate blood) and high risk (host swatting defences and predation). Although there is no variation within populations in the strategies in our model, for a given parameter set there is a single optimal strategy: individual-level variation in realized behaviour can emerge from stochastic events in the simulations. However, there were no strong divergences of behaviour between individuals, with individual trajectories being fairly similar (see electronic supplementary material, figure S9).

The hump-shaped allocation observed in tsetse in the laboratory [23] could potentially reflect an evolutionarily optimal strategy best explained by an age-dependent increase in energy intake, e.g. through experience or developing a larger gut [74], and an age-dependent increase in energetic costs, as flight, for example, may be impaired due to damage accumulation [16,17]. Relative allocation decreases with age and older females allocate less reserves to reproduction in comparison to younger females, regardless of their own reserves. This concurs with predictions of adaptive later-life reproductive restraint as a functional explanation for reproductive senescence [20], whereby maternal allocation decreases with age to reduce risks of increased mortality associated with accrued damage due to reproduction [20] or starvation with declining energy dynamics.

A caveat is that the only available data on within-individual patterns of allocation with age in tsetse are from a laboratory study with a population of flies that has been in the laboratory for many generations [23], and we cannot conclude how well our model would explain patterns in the wild. In cross-sectional studies, there is a slight increase in allocation with age, as observed at earlier ages both in the laboratory [23] and in our model, but no later-life decline [50,78]. The lack of reproductive senescence in the wild could be linked to shorter lifespans, with wild flies being more susceptible to death from starvation and predation [43,79]. A limitation of the field data is that individual tsetse cannot be tracked across their lifespan, and pregnant females are only caught during particular seasons of the year. As such, we may not be able to observe reproductive senescence in the wild, even if it occurred, due to the cross-sectional data currently available [50,78].

In summary, we provide a mechanistic explanation behind the pattern of increase-then-decrease in maternal allocation, which is driven by evolutionary constraints with age-dependent effects on energy dynamics, confirming the possibility of later-life reproductive restraint. Our model also provides a more general framework to understand

optimal reproductive allocation in iteroparous breeders. By tracking maternal allocation, maternal reserves and relative allocation, we show what strategic choices individuals make given their ecology, anywhere on the continuum from extreme capital to extreme income breeding. With our particular parameters tailored to tsetse biology, we find an income breeding strategy as we predicted given that tsetse acquire resources through feeding on protein-rich blood multiple times for each gestation cycle [50]. However, the same model could also predict a capital breeding strategy when applied to specific biology of other iteroparous breeders. Indeed, we hope that this framework inspires future models that could be fitted to long-term individual studies from wild vertebrate populations such as red deer, bison or terns [10,15] and thus ascertain the generality of our findings both in field conditions and in diverse taxonomic groups.

Data accessibility. The datasets supporting this article are available from the Dryad Digital Repository: https://doi.org/10.5061/dryad.v41ns1rxr [80].

Additional information and figures are provided in electronic supplementary material [81].

Authors' contributions. A.M.G.B.: conceptualization, data curation, formal analysis, investigation, methodology, project administration, validation, visualization, writing—original draft, writing—review and editing; A.D.H.: data curation, formal analysis, investigation, methodology, validation, visualization, writing—review and editing; M.B.B.: conceptualization, formal analysis, funding acquisition, methodology, supervision, validation, visualization, writing—review and editing; S.E.: conceptualization, formal analysis, funding acquisition, methodology, project administration, supervision, visualization, writing—review and editing.

All authors gave final approval for publication and agreed to be held accountable for the work performed therein.

Funding. This work was funded by the Biotechnology and Biological Sciences Research Council (grant no. BB/P006159/1). S.E. was also supported by a Royal Society Dorothy Hodgkin Fellowship (grant no. DH140236). A.D.H. was supported by a NERC Independent Research Fellowship (grant no. NE/L011921/1).

Competing interests. We declare we have no competing interests.

Acknowledgements. We thank John Hargrove, Glyn Vale, Lee Haines, Katherine Rock, Matthew Keeling, Steve Torr, Jennifer Lord and Robert Leyland for thoughtful discussions.

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
