## [Peer Review File · Proceedings of the Royal Society B: Biological Sciences]

Review History

RSPB-2021-1884.R0 (Original submission)

Review form: Reviewer 1

Recommendation

Major revision is needed (please make suggestions in comments)

Scientific importance: Is the manuscript an original and important contribution to its field?

Acceptable

General interest: Is the paper of sufficient general interest?

Acceptable

Quality of the paper: Is the overall quality of the paper suitable?

Good

Is the length of the paper justified?

Yes

Should the paper be seen by a specialist statistical reviewer?

No

Do you have any concerns about statistical analyses in this paper? If so, please specify them explicitly in your report.

No

It is a condition of publication that authors make their supporting data, code and materials available - either as supplementary material or hosted in an external repository. Please rate, if applicable, the supporting data on the following criteria.

Is it accessible?

Yes

Is it clear?

Yes

Is it adequate?

Yes

Do you have any ethical concerns with this paper?

No

Comments to the Author

Dear Editor,

Many organisms show a decline in reproductive success with age. This pattern, referred to as reproductive senescence, is often seen as a hump-shaped relationship between age and a fitness component (e.g., offspring size, clutch size, etc.). The evolutionary theory of aging proposes that a decline in reproductive success with age is the result of weak selection on fitness components expressed late in life. This can result in the accumulation of late-acting deleterious alleles (or alleles with antagonistic effects on early and late life fitness). Although the evolutionary theory of aging is well supported, less is known about the specific impact of ecological and energetic factors in determining the relationship between age and reproductive fitness (although extrinsic mortality is expected to be important).

In this manuscript, the authors use a dynamic state variable model to predict optimal maternal allocation patterns as a function of age, when there is stochasticity in energy dynamics and age dependent changes in factors such as feeding success and external mortality rate. The model is parameterized with data from tsetse. Using the results of their models, the authors determine what combination of factors successfully recapitulate the hump-shaped relationship between age and maternal allocation in tsetse.

The manuscript is well written and addresses a topic that is certainly of interest to the readers of Proc. R. Soc. B. I am not a theoretician, so I am not able to comment on the modeling approach. However, I have some more general comments and questions regarding the author's approach and conclusions.

1. A central theme in the evolution of senescence is that selection acting late in life is weak compared to selection acting early in life. I really struggled to see how variation in the strength of selection is incorporated in to their modeling approach (is this just through extrinsic mortality?). This was especially confusing for me because the dynamic programming approach finds the optimal strategy by beginning with the oldest age class and working backward. This seems biologically implausible, but I'm sure that I am missing something here?

2. The authors parameterize the model using data from a laboratory population of tsetse. They then determine the set of environmental conditions that generate a hump-shaped relationship between age and maternal allocation that is most similar to the relationship observed

in the lab. This approach seems to assume that the tsetse population has adapted to the lab conditions. Is there evidence for this? It's not clear from the methods whether there is evidence that the population is lab adapted or even how many generations the population has been maintained in the lab. The approach in the paper only works if the population is lab adapted.

3. The authors main conclusion is that when there is age-dependency in energy dynamics, the optimal relationship between age and maternal allocation is hump-shaped. It was never really clear to me whether the authors are proposing this as an alternative to reproductive senescence driven by relaxed selection on maternal investment late in life. I guess this gets back to my first point of confusion (related to point 1 above).

4. Line 280-281: These sentences were slightly confusing. The first sentence tells us the fraction of scenarios that generated hump-shaped allocation patterns (749/1403). The next sentence considers the "models with a negative quadratic term" that had an $R^2 > 0.7$ (35/1403). Shouldn't this fraction be 35/749? The first sentence implies that 749 of the scenarios had a negative quadratic term, so that should be the denominator in the second proportion.

5. Line 351: "have" should be "has"

6. Line 384-387: I agree that using this same approach to predict patterns of reproductive senescence in other species would be very illuminating! In my opinion, the manuscript would be of interest to a much broader audience if it included such a comparison.

7. In the abstract, the authors point out that diverse scenarios can generate hump-shaped relationships between maternal allocation and age. They then conclude that this is why so many organisms display this pattern in nature. I'm not sure that their data allow them to make such a conclusion. It is true that there appear to many routes to a hump-shaped distribution. However, without a cross-species comparison we don't know whether different species take these different routes! This conclusion would be much stronger if the authors used data from another organism to parameterize the model and explore which environmental / energetic conditions generate the pattern they are trying to explain.

8. Hargrove et al. 2011 appears twice in reference list (37 and 41) but appears to be the same paper.

Review form: Reviewer 2 (Jean-Michel Gaillard)

Recommendation

Major revision is needed (please make suggestions in comments)

Scientific importance: Is the manuscript an original and important contribution to its field?

Good

General interest: Is the paper of sufficient general interest?

Good

Quality of the paper: Is the overall quality of the paper suitable?

Good

Is the length of the paper justified?

Yes

Should the paper be seen by a specialist statistical reviewer?

No

Do you have any concerns about statistical analyses in this paper? If so, please specify them explicitly in your report.

Yes

It is a condition of publication that authors make their supporting data, code and materials available - either as supplementary material or hosted in an external repository. Please rate, if applicable, the supporting data on the following criteria.

Is it accessible?

Yes

Is it clear?

Yes

Is it adequate?

Yes

Do you have any ethical concerns with this paper?

No

Comments to the Author

This original work proposes a model of optimal allocation to reproduction in iteroparous animals with an application to the case study of tsetse flies.

I really enjoyed reading that nice piece of work that can potentially offer a solid contribution to better understand life history variation. However, I found three major problems that need to be solved.

First and most importantly, the proposed model targets the amount of reserves that mothers allocate and seems at first sight limited to extreme capital breeders (sensu Jonsson 1997 Oikos). Then, the model also includes feeding opportunities, indicating that it might be applied to income breeders. Lastly, the statement in line 142 clearly shows that the model is restricted to capital breeders: income breeders produce offspring without storing any reserves!

It is thus impossible to assess how the model proposed matches with the major axis of variation in resource allocation to reproduction: the capital-income breeder continuum. A thorough discussion about how the model should be influenced by the ranking of a given species on the capital-income breeder continuum is required. In particular, purely capital breeders should be insensitive to food restriction during the rearing period. They have information about their body reserves and can adjust their reproductive allocation to these reserves. On the contrary, purely income breeders have no information and strongly depend on resource availability during the rearing period. Moreover, storing, carrying, and handling body reserves is more costly than directly allocating energy assimilated to offspring. How the proposed model accounts for these opposed reproductive tactics and their associated constraints has to be made explicit. For instance, the conclusion that age-dependence in feeding success does not drive the shape of age-specific allocation is likely a consequence of the model structure and might not apply to income breeders.

Second, the authors often refers to « offspring quality » without defining what they mean by that. Is it offspring survival? Offspring mass? Moreover, the justification of using a Gompertz model to relate this « offspring quality » with fitness gain has to be provided.

Lastly, the use of quadratic model to describe age-specific changes in reproductive allocation is misleading. A decreasing rate of increase in reproductive performance among young and prime-aged adults with no senescence is well fitted by a quadratic model. However, to model the standard three stages expected in age-specific variation in reproduction (Emlen 1970 Ecology, which would warrant to be quoted in that paper), a threshold model is required. Threshold models are indeed more appropriate than quadratic models to assess senescence patterns (see e.g. Bermann et al. 2009 PRSB for a discussion).

Detailed comments:

- l. 28: « mortality risk », not « extrinsic mortality »
- l. 32: « mortality risk », not « extrinsic mortality »
- l. 44: « allocation », not « investment »
- l. 55: Remove « extrinsic »
- l. 58: Remove « intrinsic »
- l. 94: Remove « extrinsic »
- l. 96: « case study » instead of « an exemplar »
- l. 98: « deer », not « deers »
- l. 131; What ny refers to is missing
- l. 150: Remove « which is a form of intrinsic mortality »
- l. 151: Replace « extrinsic » with « environmentally driven »
- l. 151-152: Replace « from external factors (predation, inclement weather). » with « from predation or inclement weather. »
- l. 161: Remove « extrinsic »
- l. 198: Remove « extrinsic »
- l. 214: Remove « extrinsic »
- l. 220-221: The legend of Table 1 should be placed above the table.
- l. 227: Remove « extrinsic »
- l. 255-256: The legend of Table 2 should be placed above the table.
- l. 256: Remove « extrinsic »
- Figure 1: Display the data in addition to models in the panels
- l. 291-296: The legend of Table 3 should be placed above the table.
- l. 296: Remove « extrinsic »
- l. 324: Remove « extrinsic »
- l. 356: Remove « extrinsic »
- l. 357: Remove « extrinsic »
- l. 362: I do not undress and what a balance between intrinsic and extrinsic mortality means. In nature, both types of mortality strongly interplay, leaving no chance to tease them apart.
- l. 369-373: This is clearly an overstatement pushing too far the adaptive interpretation. Reproductive restraint has been involved for young, not old individuals (see seminal papers by Curio). From a more parsimonious viewpoint, a decreased allocation with increasing age reflects senescence.
- l. 382-384: Alternatively, the lack of reproductive senescence in the wild might be due to a shorter lifespan in the wild.

J. M. Gaillard

Review form: Reviewer 3

Recommendation

Accept with minor revision (please list in comments)

Scientific importance: Is the manuscript an original and important contribution to its field?

Excellent

General interest: Is the paper of sufficient general interest?

Excellent

Quality of the paper: Is the overall quality of the paper suitable?

Excellent

Is the length of the paper justified?

Yes

Should the paper be seen by a specialist statistical reviewer?

No

Do you have any concerns about statistical analyses in this paper? If so, please specify them explicitly in your report.

No

It is a condition of publication that authors make their supporting data, code and materials available - either as supplementary material or hosted in an external repository. Please rate, if applicable, the supporting data on the following criteria.

Is it accessible?

Yes

Is it clear?

Yes

Is it adequate?

Yes

Do you have any ethical concerns with this paper?

No

Comments to the Author

In this manuscript the authors describe an optimality model that is used to investigate the conditions that favour hump shaped patterns in maternal investment in offspring throughout the life of the mother. Overall, I think the model results are interesting and the manuscript is well written.

I therefore have only a few comments and suggestions for the authors. Some of these are just minor details, but changes would still help clarify the m.s.

My only slightly larger/more general concern is about the way you write about results from statistical tests on simulation results. While these results represent the expected population level average maternal investment with respect to age, they do not necessarily represent very well the individual choices or strategies. I have no doubt the authors agree with me on this, but I think it is necessary to clarify this in the text, especially in the discussion.

l. 91-92: do you actually investigate stochasticity in energetic costs and feeding success? The way I read your Table 2, it seems that there is no systematic investigation of differences in stochasticity? Is this specified somewhere else?

l.103-104 & l. 138: Although this is at least partially explained later in the ms, I think it would be valuable with a brief explanation of why or how there can be stochasticity in costs of flight and blood digestion (these could easily be thought of as relatively constant for a single individual)

l. 155: "fitness h", is it not better to write expected fitness here?

l. 186-187: It was unclear to me at first that this was the starting reserves at emergence, please rewrite to clarify.

l. 267-268: are you not repeating yourself here?

l. 342-343 could you link these empirical patterns more strongly to your model? Maybe through

specifying which parameters of your model might be involved? It is unclear to me how relevant these empirical results are to your model results.

1.389-396. While I do not strongly disagree with this paragraph in any way, I am not quite convinced it is a good ending as the potential implications mentioned here are quite far from the actual model results and new insight on these matters have yet come from the model.

ESM: Would it be possible to use the same name for the time axis in all the figures? Is there a difference between larviposition interval and number of reproductive cycles?

Decision letter (RSPB-2021-1884.R0)

27-Oct-2021

Dear Dr Barreaux,

Your manuscript has now been peer reviewed and the reviews have been assessed by an Associate Editor. The reviewers' comments (not including confidential comments to the Editor) and the comments from the Associate Editor are included at the end of this email for your reference. As you will see, the reviewers have raised some concerns with your manuscript and we would like to invite you to revise your manuscript to address them.

Research ethics:

Use of animals and field studies:

If your study uses animals please include details in the methods section of any approval and licences given to carry out the study and include full details of how animal welfare standards were ensured. Field studies should be conducted in accordance with local legislation; please

include details of the appropriate permission and licences that you obtained to carry out the field work.

It is a condition of publication that you make available the data and research materials supporting the results in the article. Please see our Data Sharing Policies (<https://royalsociety.org/journals/authors/author-guidelines/#data>). Datasets should be deposited in an appropriate publicly available repository and details of the associated accession number, link or DOI to the datasets must be included in the Data Accessibility section of the article (<https://royalsociety.org/journals/ethics-policies/data-sharing-mining/>). Reference(s) to datasets should also be included in the reference list of the article with DOIs (where available).

If you wish to submit your data to Dryad (<http://datadryad.org/>) and have not already done so you can submit your data via this link [http://datadryad.org/submit?journalID=RSPB&manu=\(Document not available\)](http://datadryad.org/submit?journalID=RSPB&manu=(Document%20not%20available)), which will take you to your unique entry in the Dryad repository.

Please submit a copy of your revised paper within three weeks. If we do not hear from you within this time your manuscript will be rejected. If you are unable to meet this deadline please let us know as soon as possible, as we may be able to grant a short extension.

Best wishes,
Professor Loeske Kruuk
<mailto:proceedingsb@royalsociety.org>

Associate Editor

Board Member: 1

Comments to Author:

I enjoyed reading this paper which tackles an important question of optimal allocation to reproduction in iteroparous animals. I was particularly intrigued by the lack age-dependent extrinsic mortality effects and the evidence pointing to reproductive restraint. I think this paper

can make an important contribution to the field. The paper has been reviewed by three experts that provided a number of queries and suggestions for improvement. I encourage the authors to address all of the points raised by the reviewers in detail.

Reviewer(s)' Comments to Author:

Referee: 1

Comments to the Author(s)

Dear Editor,

Many organisms show a decline in reproductive success with age. This pattern, referred to as reproductive senescence, is often seen as a hump-shaped relationship between age and a fitness component (e.g., offspring size, clutch size, etc.). The evolutionary theory of aging proposes that a decline in reproductive success with age is the result of weak selection on fitness components expressed late in life. This can result in the accumulation of late-acting deleterious alleles (or alleles with antagonistic effects on early and late life fitness). Although the evolutionary theory of aging is well supported, less is known about the specific impact of ecological and energetic factors in determining the relationship between age and reproductive fitness (although extrinsic mortality is expected to be important).

In this manuscript, the authors use a dynamic state variable model to predict optimal maternal allocation patterns as a function of age, when there is stochasticity in energy dynamics and age dependent changes in factors such as feeding success and external mortality rate. The model is parameterized with data from tsetse. Using the results of their models, the authors determine what combination of factors successfully recapitulate the hump-shaped relationship between age and maternal allocation in tsetse.

The manuscript is well written and addresses a topic that is certainly of interest to the readers of Proc. R. Soc. B. I am not a theoretician, so I am not able to comment on the modeling approach. However, I have some more general comments and questions regarding the author's approach and conclusions.

1. A central theme in the evolution of senescence is that selection acting late in life is weak compared to selection acting early in life. I really struggled to see how variation in the strength of selection is incorporated in to their modeling approach (is this just through extrinsic mortality?). This was especially confusing for me because the dynamic programming approach finds the optimal strategy by beginning with the oldest age class and working backward. This seems biologically implausible, but I'm sure that I am missing something here?
2. The authors parameterize the model using data from a laboratory population of tsetse. They then determine the set of environmental conditions that generate a hump-shaped relationship between age and maternal allocation that is most similar to the relationship observed in the lab. This approach seems to assume that the tsetse population has adapted to the lab conditions. Is there evidence for this? It's not clear from the methods whether there is evidence that the population is lab adapted or even how many generations the population has been maintained in the lab. The approach in the paper only works if the population is lab adapted.
3. The authors main conclusion is that when there is age-dependency in energy dynamics, the optimal relationship between age and maternal allocation is hump-shaped. It was never really clear to me whether the authors are proposing this as an alternative to reproductive senescence driven by relaxed selection on maternal investment late in life. I guess this gets back to my first point of confusion (related to point 1 above).
4. Line 280-281: These sentences were slightly confusing. The first sentence tells us the fraction of scenarios that generated hump-shaped allocation patterns (749/1403). The next sentence considers the "models with a negative quadratic term" that had an $R^2 > 0.7$ (35/1403). Shouldn't this fraction be 35/749? The first sentence implies that 749 of the scenarios had a negative quadratic term, so that should be the denominator in the second proportion.

5. Line 351: “have” should be “has”

6. Line 384-387: I agree that using this same approach to predict patterns of reproductive senescence in other species would be very illuminating! In my opinion, the manuscript would be of interest to a much broader audience if it included such a comparison.

7. In the abstract, the authors point out that diverse scenarios can generate hump-shaped relationships between maternal allocation and age. They then conclude that this is why so many organisms display this pattern in nature. I’m not sure that their data allow them to make such a conclusion. It is true that there appear to many routes to a hump-shaped distribution. However, without a cross-species comparison we don’t know whether different species take these different routes! This conclusion would be much stronger if the authors used data from another organism to parameterize the model and explore which environmental / energetic conditions generate the pattern they are trying to explain.

8. Hargrove et al. 2011 appears twice in reference list (37 and 41) but appears to be the same paper.

Referee: 2

Comments to the Author(s)

This original work proposes a model of optimal allocation to reproduction in iteroparous animals with an application to the case study of tsetse flies.

I really enjoyed reading that nice piece of work that can potentially offer a solid contribution to better understand life history variation. However, I found three major problems that need to be solved.

First and most importantly, the proposed model targets the amount of reserves that mothers allocate and seems at first sight limited to extreme capital breeders (*sensu* Jonsson 1997 *Oikos*). Then, the model also includes feeding opportunities, indicating that it might be applied to income breeders. Lastly, the statement in line 142 clearly shows that the model is restricted to capital breeders: income breeders produce offspring without storing any reserves!

It is thus impossible to assess how the model proposed matches with the major axis of variation in resource allocation to reproduction: the capital-income breeder continuum. A thorough discussion about how the model should be influenced by the ranking of a given species on the capital-income breeder continuum is required. In particular, purely capital breeders should be insensitive to food restriction during the rearing period. They have information about their body reserves and can adjust their reproductive allocation to these reserves. On the contrary, purely income breeders have no information and strongly depend on resource availability during the rearing period. Moreover, storing, carrying, and handling body reserves is more costly than directly allocating energy assimilated to offspring. How the proposed model accounts for these opposed reproductive tactics and their associated constraints has to be made explicit. For instance, the conclusion that age-dependence in feeding success does not drive the shape of age-specific allocation is likely a consequence of the model structure and might not apply to income breeders.

Second, the authors often refers to « offspring quality » without defining what they mean by that. Is it offspring survival? Offspring mass? Moreover, the justification of using a Gompertz model to relate this « offspring quality » with fitness gain has to be provided.

Lastly, the use of quadratic model to describe age-specific changes in reproductive allocation is misleading. A decreasing rate of increase in reproductive performance among young and prime-aged adults with no senescence is well fitted by a quadratic model. However, to model the standard three stages expected in age-specific variation in reproduction (Emlen 1970 *Ecology*, which would warrant to be quoted in that paper), a threshold model is required. Threshold

models are indeed more appropriate than quadratic models to assess senescence patterns (see e.g. Bermann et al. 2009 PRSB for a discussion).

Detailed comments:

- l. 28: « mortality risk », not « extrinsic mortality »
- l. 32: « mortality risk », not « extrinsic mortality »
- l. 44: « allocation », not « investment »
- l. 55: Remove « extrinsic »
- l. 58: Remove « intrinsic »
- l. 94: Remove « extrinsic »
- l. 96: « case study » instead of « an exemplar »
- l. 98: « deer », not « deers »
- l. 131; What ny refers to is missing
- l. 150: Remove « which is a form of intrinsic mortality »
- l. 151: Replace « extrinsic » with « environmentally driven »
- l. 151-152: Replace « from external factors (predation, inclement weather). » with « from predation or inclement weather. »
- l. 161: Remove « extrinsic »
- l. 198: Remove « extrinsic »
- l. 214: Remove « extrinsic »
- l. 220-221: The legend of Table 1 should be placed above the table.
- l. 227: Remove « extrinsic »
- l. 255-256: The legend of Table 2 should be placed above the table.
- l. 256: Remove « extrinsic »
- Figure 1: Display the data in addition to models in the panels
- l. 291-296: The legend of Table 3 should be placed above the table.
- l. 296: Remove « extrinsic »
- l. 324: Remove « extrinsic »
- l. 356: Remove « extrinsic »
- l. 357: Remove « extrinsic »
- l. 362: I do not undress and what a balance between intrinsic and extrinsic mortality means. In nature, both types of mortality strongly interplay, leaving no chance to tease them apart.
- l. 369-373: This is clearly an overstatement pushing too far the adaptive interpretation. Reproductive restraint has been involved for young, not old individuals (see seminal papers by Curio). From a more parsimonious viewpoint, a decreased allocation with increasing age reflects senescence.
- l. 382-384: Alternatively, the lack of reproductive senescence in the wild might be due to a shorter lifespan in the wild.

J. M. Gaillard

Referee: 3

Comments to the Author(s)

In this manuscript the authors describe an optimality model that is used to investigate the conditions that favour hump shaped patterns in maternal investment in offspring throughout the life of the mother. Overall, I think the model results are interesting and the manuscript is well written.

I therefore have only a few comments and suggestions for the authors. Some of these are just minor details, but changes would still help clarify the m.s.

My only slightly larger/more general concern is about the way you write about results from statistical tests on simulation results. While these results represent the expected population level average maternal investment with respect to age, they do not necessarily represent very well the individual choices or strategies. I have no doubt the authors agree with me on this, but I think it is necessary to clarify this in the text, especially in the discussion.

l. 91-92: do you actually investigate stochasticity in energetic costs and feeding success? The way I read your Table 2, it seems that there is no systematic investigation of differences in stochasticity? Is this specified somewhere else?

l.103-104 & l. 138: Although this is at least partially explained later in the ms, I think it would be valuable with a brief explanation of why or how there can be stochasticity in costs of flight and blood digestion (these could easily be thought of as relatively constant for a single individual)

l. 155: "fitness h", is it not better to write expected fitness here?

l. 186-187: It was unclear to me at first that this was the starting reserves at emergence, please rewrite to clarify.

l. 267-268: are you not repeating yourself here?

l. 342-343 could you link these empirical patterns more strongly to your model? Maybe through specifying which parameters of your model might be involved? It is unclear to me how relevant these empirical results are to your model results.

l.389-396. While I do not strongly disagree with this paragraph in any way, I am not quite convinced it is a good ending as the potential implications mentioned here are quite far from the actual model results and new insight on these matters have yet come from the model.

ESM: Would it be possible to use the same name for the time axis in all the figures? Is there a difference between larviposition interval and number of reproductive cycles?

Author's Response to Decision Letter for (RSPB-2021-1884.R0)

See Appendix A.

RSPB-2021-1884.R1 (Revision)

Review form: Reviewer 1

Recommendation

Major revision is needed (please make suggestions in comments)

Scientific importance: Is the manuscript an original and important contribution to its field?

Good

General interest: Is the paper of sufficient general interest?

Acceptable

Quality of the paper: Is the overall quality of the paper suitable?

Good

Is the length of the paper justified?

Yes

Should the paper be seen by a specialist statistical reviewer?

No

Do you have any concerns about statistical analyses in this paper? If so, please specify them explicitly in your report.

No

It is a condition of publication that authors make their supporting data, code and materials available - either as supplementary material or hosted in an external repository. Please rate, if applicable, the supporting data on the following criteria.

Is it accessible?

Yes

Is it clear?

Yes

Is it adequate?

Yes

Do you have any ethical concerns with this paper?

No

Comments to the Author

Dear Editor,

I reviewed a previous version of this manuscript and the authors have done an excellent job responding to my previous comments of the other reviewers. Nevertheless, I have a few comments regarding the revised manuscript. The first two issues are relatively minor. The third is more substantial.

1. Line 22, line 388 (and elsewhere): There are two major explanations for the evolution of senescence: optimality and mutation pressure (e.g., Partridge and Barton 1993). The optimality explanation posits that ageing could evolve as part of an optimal life history in which there is an antagonism between performance early in life and performance late in life. In contrast, mutation pressure could lead to ageing because selection against late-acting deleterious mutations will be weak. These two explanations cannot be distinguished based upon the relationship between parental allocation and parental age. Thus, I do not think it is correct to say that "optimal" allocation is hump shaped across ages in diverse taxa. The hump-shaped pattern may be widespread, but in most cases we do not know whether the shape of this curve can be explained by the optimality hypothesis or the mutation pressure hypothesis.

2. Line 51: genes should be alleles.

3. Line 80-87 and 236-238: I have a hard time reconciling these two sections. In the first passage, the authors acknowledge that offspring quality may decline with parental age, and as a consequence later-born offspring might have lower reproductive value to parents than earlier-born offspring. The authors suggest that this pattern is not accounted for in current models of senescence, which focus on fecundity and not offspring quality. However, in the second passage the authors state that they are assuming that the reproductive value of earlier-born and later-born offspring is the same. I understand that the goal is to examine whether maternal allocation strategies can generate hump-shaped allocation curves without invoking a decline in the strength of selection with age. However, it is not clear whether / how the model predictions will change if the assumption that all offspring contribute equally to parental fitness is relaxed.

Review form: Reviewer 2

Recommendation

Accept with minor revision (please list in comments)

Scientific importance: Is the manuscript an original and important contribution to its field?
Excellent

General interest: Is the paper of sufficient general interest?
Excellent

Quality of the paper: Is the overall quality of the paper suitable?
Excellent

Is the length of the paper justified?
Yes

Should the paper be seen by a specialist statistical reviewer?
No

Do you have any concerns about statistical analyses in this paper? If so, please specify them explicitly in your report.
No

It is a condition of publication that authors make their supporting data, code and materials available - either as supplementary material or hosted in an external repository. Please rate, if applicable, the supporting data on the following criteria.

Is it accessible?
Yes

Is it clear?
Yes

Is it adequate?
Yes

Do you have any ethical concerns with this paper?
No

Comments to the Author

I warmly thank the authors for having provided both detailed responses to comments and a carefully-revised manuscript. I was really convinced by the changes performed (in particular the link between the authors' model and the capital-income breeder continuum). However, I still have two concerns about this paper, which should be easy to solve.

First, the two assumptions the authors clearly stated in their responses to the first reviewer's comments should be presented and discussed more thoroughly in the paper itself. The « as » in 1.322 in the version with tracked changes should be replaced with « as we assume ». Indeed, there appear to be quite strong assumptions because in the real world (1) the timing of reproduction throughout the lifetime matters a lot in terms of fitness but in case the population is stationary, and (2) mutations should occur soon or later.

Second, I still disagree with the use of quadratic models to infer an initial increase of reproductive performance until a peak or plateau followed by a decrease of reproductive performance. To illustrate my concern, consider the following age-specific trajectory of performance (rp):

age$\in\{1,2,3,4,5,6,7,8,9,10,11\}$

rp$\in\{0.01,0.05,0.20,0.60,1.2,1.5,1.6,1.7,1.75,1.75,1.75\}$

This rp trajectory is characterized by a continuous increase of reproduction with age until a plateau is reached. There is no evidence of any reproductive senescence from this trajectory. Now fit a quadratic model:

$\text{mod}\&\text{lt};\text{lm}(\text{rp} \sim \text{age} + \text{I}(\text{age}^2))$

and look at the estimates:

Coefficients:

	Estimate	Std. Error	t value	Pr(> t)
(Intercept)	-0.708182	0.206638	-3.427	0.008992 **
age	0.469056	0.079144	5.927	0.000351 ***
I(age^2)	-0.021853	0.006424	-3.402	0.009333 **

The outcome is a statistically significant negative quadratic coefficient in absence of reproductive senescence. This clearly demonstrates that a negative quadratic term cannot be interpreted in terms of decreasing performance in late life. Although additional models ran by the authors convinced me that their findings are robust, I would ask the authors to add an explicit statement about the caution required when interpreting quadratic parameters in terms of senescence to avoid the pitfall mentioned just above.

J.M. Gaillard

Author's Response to Decision Letter for (RSPB-2021-1884.R1)

See Appendix B.

Decision letter (RSPB-2021-1884.R2)

18-Jan-2022

Dear Dr Barreaux

I am pleased to inform you that your manuscript entitled "Incorporating effects of age on energy dynamics predicts non-linear maternal allocation patterns in iteroparous animals" has been accepted for publication in Proceedings B.

Data Accessibility section

Open Access

Paper charges

Sincerely,

Proceedings B

Appendix A

University of
BRISTOL

Dr Antoine Barreaux
Research Associate

School of Biological Sciences, Life Sciences Building
email:antoine.barreaux@bristol.ac.uk

08/12/21

Dear Prof Kruuk

We are resubmitting our research manuscript entitled "*Incorporating effects of age on energy dynamics predicts non-linear maternal allocation patterns in iteroparous animals*" for consideration for publication in *Proceedings of the Royal Society B*. We thank you for considering our manuscript and for the opportunity to resubmit.

We are grateful to the associate editor and reviewers for the positive reviews and helpful comments that we feel have improved the manuscript. We have significantly changed sections of this manuscript in light of each reviewer's comments and have incorporated all the stylistic and grammatical corrections suggested. Our detailed responses are provided below.

We describe original work that is not being considered for publication in any other journal. We declare having no competing interests. All the authors gave final approval for publication and submission to *Proceedings B* of the final manuscript.

We appreciate your time and are looking forward to your response.

Best regards,

Antoine Barreaux, on behalf of all authors

Associate Editor

Board Member: 1

Comments to Author:

I enjoyed reading this paper which tackles an important question of optimal allocation to reproduction in iteroparous animals. I was particularly intrigued by the lack age-dependent extrinsic mortality effects and the evidence pointing to reproductive restraint. I think this paper can make an important contribution to the field. The paper has been reviewed by three experts that provided a number of queries and suggestions for improvement. I encourage the authors to address all of the points raised by the reviewers in detail.

RESPONSE: We thank the associate editor for their time, positive comments, and consideration for our manuscript. We have made sure to answer all the helpful queries and suggestions made by the three referees, and to address all the points raised. Please find our detailed answer below.

Referee: 1

Many organisms show a decline in reproductive success with age. This pattern, referred to as reproductive senescence, is often seen as a hump-shaped relationship between age and a fitness component (e.g., offspring size, clutch size, etc.). The evolutionary theory of aging proposes that a decline in reproductive success with age is the result of weak selection on fitness components expressed late in life. This can result in the accumulation of late-acting deleterious alleles (or alleles with antagonistic effects on early and late life fitness). Although the evolutionary theory of aging is well supported, less is known about the specific impact of ecological and energetic factors in determining the relationship between age and reproductive fitness (although extrinsic mortality is expected to be important).

In this manuscript, the authors use a dynamic state variable model to predict optimal maternal allocation patterns as a function of age, when there is stochasticity in energy dynamics and age dependent changes in factors such as feeding success and external mortality rate. The model is parameterized with data from tsetse. Using the results of their models, the authors determine what combination of factors successfully recapitulate the hump-shaped relationship between age and maternal allocation in tsetse.

The manuscript is well written and addresses a topic that is certainly of interest to the readers of Proc. R. Soc. B. I am not a theoretician, so I am not able to comment on the modeling approach. However, I have some more general comments and questions regarding the author's approach and conclusions.

RESPONSE: We thank the referee for their interest in our manuscript. We appreciate the helpful general comments and suggestions on our approach and conclusions.

1. A central theme in the evolution of senescence is that selection acting late in life is weak compared to selection acting early in life. I really struggled to see how variation in the strength of selection is incorporated in to their modeling approach (is this just through extrinsic mortality?). This was especially confusing for me because the dynamic programming approach finds the optimal strategy by beginning with the oldest age class and working backward. This seems biologically implausible, but I'm sure that I am missing something here?

RESPONSE: While a decline in the strength of selection in later ages is indeed a central theme in evolutionary explanations for actuarial senescence, our intention was to identify if and when observed patterns of reproductive senescence arose through optimal maternal allocation. In our model, there is no forced decline in selective pressure later in life because all offspring are equally valuable at all ages (offspring quality depends on maternal allocation itself) and there are no mutations. Mothers may vary allocation of resources to offspring with age, which will then result in offspring of different quality and may lead to reproductive senescence if offspring quality decreases with maternal age. In the baseline model, the input parameters are age independent. We compare outputs of this model to other scenarios which include age-dependence in energy dynamics and environmentally-driven mortality ("extrinsic mortality") to study how these evolutionary constraints affect optimal allocation and whether they lead to reproductive senescence.

We have added the following clarification about how our model fits in the context of declining selection as an explanation for senescence, at 235-238:

"Our aim is to identify whether observed senescence can arise from optimal maternal allocation. As such, we do not impose a decline in selection in later life as all offspring are equally valuable at all ages (for a given maternal allocation), and there are no mutations."

To address the point about dynamic programming, we note that the approach of dynamic programming is not intended to mimic any biological process, but to find an optimal strategy. Decisions (e.g. amount of resources to allocate to an offspring) are assumed to depend on the animal's current state, regardless of how this state was reached. However, the current decision of an animal affects its future state and so future decisions. When finding the optimal current decision, it is therefore necessary to know what the future states (and decisions) will be, so we have to work backwards. This is a standard approach in dynamic programming in general introduced by Bellman in the 1950s [1] and widely applied to behavioural ecology in the 1980s and 1990s [2,3]. We do not claim that animals themselves follow such complex calculations, but follow evolved rules of thumb that generate behaviour similar to the strategy [4].

2. The authors parameterize the model using data from a laboratory population of tsetse. They then determine the set of environmental conditions that generate a hump-shaped relationship between age and maternal allocation that is most similar to the relationship observed in the lab. This approach seems to assume that the tsetse population has adapted to the lab conditions. Is there evidence for this? It's not clear from the methods whether there is evidence that the population is lab adapted or even how many generations the population has been maintained in the lab. The approach in the paper only works if the population is lab adapted.

RESPONSE: We agree with the referee that there are potential limitations when relying too heavily on a single study from a long-standing laboratory population, and we cannot rule out effects of inbreeding or inadvertent selection. We do not assume, however, that the population has adapted to the lab, rather we ask if this laboratory population might follow the evolved strategy of field flies. This is because our assumptions are based on field conditions, including uncertain blood supply and environmentally-driven mortality. Ideally, we would fit our model to within-individual allocation data from field flies but owing to current sampling limitations, such data are not available.

We clarified this caveat in the discussion at lines 376-385:

“A caveat is that the only available data on within-individual patterns of allocation with age in tsetse is from a laboratory study with a population of flies that has been in the laboratory many generations [24], and we cannot conclude how well our model would explain patterns in the wild. In cross-sectional studies, there is a slight increase in allocation with age, as observed at earlier ages both in the laboratory [24] and in our model, but no later-life decline [51,80]. The lack of reproductive senescence in the wild could be linked to shorter lifespans, with wild flies being more susceptible to death from starvation and predation [44,81]. A limitation of the field data is that individual tsetse cannot be tracked across their lifespan and pregnant females are only caught during particular seasons of the year. As such, we may not be able to observe reproductive senescence in the wild, even if it occurred, due to the cross-sectional data currently available [51,80].”

3. The authors main conclusion is that when there is age-dependency in energy dynamics, the optimal relationship between age and maternal allocation is hump-shaped. It was never really clear to me whether the authors are proposing this as an alternative to reproductive senescence driven by relaxed selection on maternal investment late in life. I guess this gets back to my first point of confusion (related to point 1 above).

RESPONSE: Yes, in our study we ask whether reproductive senescence can be predicted without invoking a mutation-selection balance. As explained above, there is no forced changing selective pressure later in life in our model because all offspring are equally valuable at all ages and there are no mutations. We here predict reproductive senescence with our model from an optimal allocation perspective. This concurs with the findings of McNamara et al. [5] showing how reproductive senescence can be condition-dependent, with later-life reproductive restraint being due to damage accumulation. Here, we show that an evolutionary explanation for reproductive senescence could be age-dependent changes in resource availability and energetic costs (energy dynamics) and not variation in environmentally-driven mortality – at least in a tsetse-like organism.

We clarified lines 369-374:

“Relative allocation decreases with age and older females allocate less reserves to reproduction in comparison to younger females, regardless of their own reserves. This concurs with predictions of adaptive later-life reproductive restraint as a functional explanation for reproductive senescence [20], whereby maternal allocation decreases with age to reduce risks of increased mortality associated with accrued damage due to reproduction [20] or starvation with declining energy dynamics.”

We also added lines 387-389:

“In summary, we provide a mechanistic explanation behind the pattern of increase-then-decrease in optimal allocation which is driven by evolutionary constraints with age-dependent effects on energy dynamics, confirming the possibility of later-life reproductive restraint.”

4. Line 280-281: These sentences were slightly confusing. The first sentence tells us the fraction of scenarios that generated hump-shaped allocation patterns (749/1403). The next sentence considers the “models with a negative quadratic term” that had an $R^2 > 0.7$ (35/1403). Shouldn't this fraction be 35/749? The first sentence implies that 749 of the scenarios had a negative quadratic term, so that should be the denominator in the second proportion.

RESPONSE: This is a good point. To avoid confusion about percentage values, we now report the absolute number in the second statement.

Lines 277-279:

"We predicted a non-linear (hump-shaped) pattern of allocation in 53% (749/1403) of the scenarios evaluated. 35 of these quadratic downward scenarios fit the simulated data with a conditional pseudo R^2 value above 0.7 (Table 3, Figure 2)."

5. Line 351: "have" should be "has"

RESPONSE: Corrected.

6. Line 384-387: I agree that using this same approach to predict patterns of reproductive senescence in other species would be very illuminating! In my opinion, the manuscript would be of interest to a much broader audience if it included such a comparison.

RESPONSE: We agree that it could be interesting and insightful to apply our model to data from a range of species, yet this is beyond the scope of our current study. Here, we predict reproductive senescence from an optimal allocation perspective. We believe our model set-up to be already quite general as it allows the optimal strategy for iteroparous animals to be anywhere on the continuum from extreme capital to extreme income breeding (see also our reply to Referee 2). We are indeed concerned with predicting where animals will lie on the continuum given their ecology (see [6]). Besides, given the range of parameters we explored, we were able to investigate a range of age-dependent allocation patterns (see Fig. 2 for those with a quadratic downward fit) without too many constraints. Finally, we note that, while we fit our model to tsetse data, it remains a general approach as the age-dependence in energy-related traits underlying the hump-shaped pattern are observed across numerous animal systems. For example, age-dependence in energy intake or feeding success is likely to be commonly observed given that foraging efficiency improves with age in many taxa. We appreciate that interesting contrasts may be raised by tailoring our model to the biology of other iteroparous systems across the phylogenetic tree, particularly those with detailed field data on age-specific reproductive allocation (e.g. long-term studies of individually marked vertebrates), and we hope our model will inspire such future work.

We added lines 120-123:

"The model set-up allows the optimal strategy to be anywhere on the continuum from extreme capital (females build up stored reserves which are used for reproduction) to extreme income breeding (females do not store reserves across breeding events and use only those acquired during feeding), given their ecology [50]."

We clarified at lines 329-337:

"These scenarios confirm the impact on allocation of gains in experience in breeding and in acquiring food [7,15,23] and increasing energetic costs across the lifespan because of damage accumulation [7,15,19,23]. The evidence for age-dependence in such traits is wide-ranging across systems from a decrease in energetic costs with an improved lactation ability (e.g. seals [25]), an improved energy transfer efficiency (rats [77]), or reduced metabolic requirements post maturation (tsetse [60,61,69]), to an increase in energy intake with an improved mobility post maturation (tsetse [60,61,70]), or a decrease in energy intake later in life because of gut deterioration (Drosophila [78]) or other physiological deteriorations. Such evidence could explain why these non-linear patterns of maternal allocation are found across diverse taxonomic groups."

We added lines 389-399:

"Our model also provides a more general framework to understand optimal reproductive allocation in iteroparous breeders. By tracking maternal allocation, maternal reserves, and relative allocation we show what strategic choices individuals make given their ecology, anywhere on the continuum from extreme capital to extreme income breeding. With our particular parameters tailored to tsetse biology, we find an income breeding strategy as we predicted given that tsetse acquire resources through feeding on protein-rich blood multiple times for each gestation cycle [51]. However, the same model could also predict a capital breeding strategy when applied to specific biology of other iteroparous breeders. Indeed, we hope that this framework inspires future models which could be fitted to long-term individual studies from wild

vertebrate populations such as red deer, bison, or terns [10, 15] and thus ascertain the generality of our findings both in field conditions and in diverse taxonomic groups.”

7. In the abstract, the authors point out that diverse scenarios can generate hump-shaped relationships between maternal allocation and age. They then conclude that this is why so many organisms display this pattern in nature. I'm not sure that their data allow them to make such a conclusion. It is true that there appear to be many routes to a hump-shaped distribution. However, without a cross-species comparison we don't know whether different species take these different routes! This conclusion would be much stronger if the authors used data from another organism to parameterize the model and explore which environmental / energetic conditions generate the pattern they are trying to explain.

RESPONSE: It's true that a large parameter space doesn't necessarily equate to a large number of species. However, we have linked empirical patterns from different species more specifically to our model by discussing the wide-ranging examples which apply to the key parameters involved, i.e. species which have age-dependence in specific traits linked to energy dynamics (see response to point 6).

As explained in our response to point 6, we believe it to be beyond the scope of this current study to use data from other organisms. Here, we predict reproductive senescence from an optimal allocation perspective, showing that an evolutionary and mechanistic explanation could be age-dependent changes in energy dynamics. We believe our current model set-up to be already quite general as it allows the optimal strategy for iteroparous animals to be anywhere on the continuum from extreme capital to extreme income breeding.

8. Hargrove et al. 2011 appears twice in reference list (37 and 41) but appears to be the same paper.

RESPONSE: Corrected.

Referee: 2

Comments to the Author(s)

This original work proposes a model of optimal allocation to reproduction in iteroparous animals with an application to the case study of tsetse flies.

I really enjoyed reading that nice piece of work that can potentially offer a solid contribution to better understand life history variation. However, I found three major problems that need to be solved.

RESPONSE: Thank you for positive comments and suggestions.

First and most importantly, the proposed model targets the amount of reserves that mothers allocate and seems at first sight limited to extreme capital breeders (sensu Jonsson 1997 Oikos). Then, the model also includes feeding opportunities, indicating that it might be applied to income breeders. Lastly, the statement in line 142 clearly shows that the model is restricted to capital breeders: income breeders produce offspring without storing any reserves!

It is thus impossible to assess how the model proposed matches with the major axis of variation in resource allocation to reproduction: the capital-income breeder continuum. A thorough discussion about how the model should be influenced by the ranking of a given species on the capital-income breeder continuum is required. In particular, purely capital breeders should be insensitive to food restriction during the rearing period. They have information about their body reserves and can adjust their reproductive allocation to these reserves. On the contrary, purely income breeders have no information and strongly depend on resource availability during the rearing period. Moreover, storing, carrying, and handling body reserves is more costly than directly allocating energy assimilated to offspring. How the proposed model accounts for these opposed reproductive tactics and their associated constraints has to be made explicit. For instance, the conclusion that age-dependence in feeding success does not drive the shape of age-specific allocation is likely a consequence of the model structure and might not apply to income breeders.

RESPONSE: We appreciate that setting our model in the context of the capital-income breeding continuum is an important point. While our manuscript may not have been explicitly clear on this point, our model does indeed allow us to predict where animals will lie on the continuum given their ecology (see [6]). One of the main model outputs is the amount of reserves females store versus use up for each offspring, thus allowing us to predict the optimal strategy anywhere along the continuum from extreme capital (rely only on stored reserves) to income (rely only on resources from food intake) breeding.

What emerges in our baseline tsetse model is almost pure income breeding: females have a minimum reserve level and every period use everything they have above this. We emphasise that this is an output rather than an assumption of the model. We have previously hypothesised that tsetse are income breeders due to two lines of evidence: first, female acquire resources through feeding on protein-rich blood multiple times for each gestation cycle, and, second, mothers delay allocation of resources to offspring until late into gestation. This latter strategy may be adaptive if it benefits females to allocate resources significantly towards offspring growth only at the point during gestation when they are certain to have sufficient reserves available to produce a viable offspring, otherwise they can abort that offspring [7].

We clarified lines 146-147 that individuals have the choice to allocate resources or not and we replaced “reserves” by “resources”:

“If the optimal decision is not to allocate any resources ($M_t^ = 0$), then no offspring can be produced. If resources are allocated ($M_t^* > 0$), then a juvenile offspring is produced.”*

As explained in our response to Referee 1 point 6 above, we have clarified our text, in the introduction at lines 120-123 and in the discussion at lines 389-399, to incorporate the point about capital versus income breeding more explicitly.

Second, the authors often refers to « offspring quality » without defining what they mean by that. Is it offspring survival? Offspring mass? Moreover, the justification of using a Gompertz model to relate this « offspring quality » with fitness gain has to be provided.

RESPONSE: In our model, offspring quality is a measure of an offspring’s expected reproductive success during its lifetime given its energy reserves at the start of adulthood. It is the value of the offspring’s life defined as the immediate gain in fitness for a mother for a given offspring based on the expected fitness of the offspring upon reaching adulthood.

We clarified this at lines 174-176:

“The immediate gain in fitness f is a function of offspring quality (i.e. the expected reproductive success during the offspring’s lifetime given its energy reserves at the start of adulthood) and depends on maternal allocation M_t minus the energy needed to survive the non-feeding phase p .”

The Gompertz assumption has no effect on our model predictions, since it is replaced by the outcome of the previous iteration over several backwards iterations (i.e. several generations). We use this Gompertz assumption as it provides an initial starting assumption about the value of offspring, and choose a sensible shape given the assumed selective mortality of smaller offspring. Since we iterate to convergence of the offspring value function, the actual shape of the initial function has no effect on our predictions. We apologise that this was unclear, and we have now clarified this in the ESM lines 20-25:

“For the immediate gain in fitness f (figure S1), the Gompertz assumption has no effect on the predictions, since it is replaced by the outcome of the previous iteration over several backwards iterations (i.e. several generations). We have to start with some assumption about the value of offspring, and choose a sensible shape given the assumed selective mortality of smaller offspring. Since we iterate to convergence of the offspring value function, the actual shape of the initial function has no effect on model predictions.”

Lastly, the use of quadratic model to describe age-specific changes in reproductive allocation is misleading. A decreasing rate of increase in reproductive performance among young and prime-aged adults with no senescence is well fitted by a quadratic model. However, to model the standard three stages expected in age-specific variation in reproduction (Emlen 1970 Ecology, which would warrant to be quoted in that paper), a threshold model is required. Threshold models are indeed more appropriate than quadratic models to assess senescence patterns (see e.g. Bermann et al. 2009 PRSB for a

discussion).

RESPONSE:

We thank the referee for their suggestion, and we have quoted the Emlen [8] and Berman [9] references in our manuscript.

We clarified lines 64-66 adding the Emlen reference:

“To our knowledge, only one study predicted an increase and decrease in fecundity as a by-product of natural selection [26] and few – if any – theoretical studies predict this lifetime reproductive resources allocation pattern, from an initial increase to a later-life decline.”

We added the Berman reference at lines 49-50:

“In many systems, maternal allocation tends to decline with age, termed reproductive senescence [10,15,16].”

And at line 99:

“similar to long-lived vertebrates [10,15,16]”

However, we respectfully disagree with the referee about a threshold model (a 3 splines threshold model for example) being required here. These splines are low order polynomials that use knot/data breakpoints to determine relationships to data and are similar to our quadratic approach. We here explain the increase-then-decrease specifically, using the quadratic model fitting with a negative quadratic term to find model scenarios with the senescence part as well as the initial increase. Our quadratic model captures increase-then-decrease patterns without assuming a straight plateau, as organisms do not necessarily show such hard switches – reproductive competence and senescence can be quite gradual.

Besides, we acknowledge the willingness to fit the best model when considering specific empirical data, as is the case for the threshold model fit to southern fulmar data in the Berman study [9]. However, in our previous study on reproductive allocation in laboratory tsetse [10], we also fitted (in addition to a quadratic model) a model assuming a relationship between offspring quality and the logarithm of maternal age, as in Berman et al. [9], as well as more flexible generalized additive models. Neither approaches resulted in an improved fit to the data compared to the quadratic fit. This brings further support to quadratic models being an appropriate and similar approach to threshold models to fit the entire life-long allocation pattern in our study, from the increase to the decrease in allocation, both from a general and tsetse-specific perspective. We accept that the model fitting algorithm may differ when applying our framework to other studies in different systems (see response to Referee 1 point 6).

Detailed comments:

l. 28: « mortality risk », not « extrinsic mortality »

l. 32: « mortality risk », not « extrinsic mortality »

l. 44: « allocation », not « investment »

l. 55: Remove « extrinsic »

l. 58: Remove « intrinsic »

l. 94: Remove « extrinsic »

l. 96: « case study » instead of « an exemplar »

l. 98: « deer », not « deers »

l. 131; What ny refers to is missing

l. 150: Remove « which is a form of intrinsic mortality »

l. 151: Replace « extrinsic » with « environmentally driven »

l. 151-152: Replace « from external factors (predation, inclement weather). » with « from predation or inclement weather. »

l. 161: Remove « extrinsic »

l. 198: Remove « extrinsic »

l. 214: Remove « extrinsic »

l. 220-221: The legend of Table 1 should be placed above the table.

l. 227: Remove « extrinsic »

l. 255-256: The legend of Table 2 should be placed above the table.

l. 256: Remove « extrinsic »

Figure 1: Display the data in addition to models in the panels

I. 291-296: The legend of Table 3 should be placed above the table.

I. 296: Remove « extrinsic »

I. 324: Remove « extrinsic »

I. 356: Remove « extrinsic »

I. 357: Remove « extrinsic »

RESPONSE: Thank you – we have made all these minor changes. We have removed “extrinsic” and intrinsic where needed and have replaced “extrinsic mortality” by “environmentally-driven mortality” to clarify (we could not just say ‘mortality’ as this is not the only source of mortality, given that individuals can starve to death)

For “ny”, we have added an explanation at lines 136-137:

“ny is the total energy intake (y units of energy per successful feeding attempt, n times per period)”

We are not sure what the referee meant by displaying the data for Figure 1 in addition to models in the panels. The simulated data is already present on the panels as we show the average maternal allocation, or average relative allocation, or average maternal reserves of 1000 mothers for 12 reproductive cycles (x-axis). An example of the raw data for the simulation of the 1000 mothers can be seen in the ESM “Figure S9. Individual forward simulation allocation data for 1000 mothers for the selected hump-shaped scenario”. However, we do not think it would help the readers understand the figure if we added these 1000 lines of raw simulated data. If the referee is instead mentioning the laboratory data from Lord et al. [10], then the quadratic fit of the tsetse laboratory data is the dashed black line visible in Figure.2. However, we do not have any empirical data for maternal reserves or relative allocation over time and we would only be able to add the empirical data for maternal allocation in panel a of figure.1. We feel such partial presentation of data would confuse the reader and decided not to modify the figure.

I. 362: I do not understand and what a balance between intrinsic and extrinsic mortality means. In nature, both types of mortality strongly interplay, leaving no chance to tease them apart.

RESPONSE: Point taken, this was indeed unclear. We have modified lines 351-354:

“We also do not consider damage accumulation depending on maintenance, which would increase the risk of damage-associated mortality with age, and potentially shift the allocation trade-off towards increased reproduction. “

I. 369-373: This is clearly an overstatement pushing too far the adaptive interpretation. Reproductive restraint has been involved for young, not old individuals (see seminal papers by Curio). From a more parsimonious viewpoint, a decreased allocation with increasing age reflects senescence.

RESPONSE: We agree with the reviewer that we here predict reproductive senescence and with our model we are looking for an evolutionary explanation for senescence from a functional perspective. We are tracking maternal reserves and relative maternal allocation, which enables us to show that individuals make the strategic choice when getting older to allocate less to reproduction despite having as much reserves as younger females. We do not agree that this pushes the adaptive interpretation too far, rather we argue that this pattern of optimal allocation is driven by evolutionary constraints with age-dependent effects on energy dynamics. This concurs with the findings of McNamara et al. [5] showing how reproductive senescence can be condition-dependent. In their model, later-life reproductive restraint is driven by reproduction damage and an increase in associated mortality risk. In our model, such later-life ‘restraint’ is driven by age-dependent energy acquisition and energetic costs, which increases the risk of condition dependent-death (starvation) when investing as much as younger mothers.

We clarified this at lines 369-374:

“Relative allocation decreases with age and older females allocate less reserves to reproduction in comparison to younger females, regardless of their own reserves. This concurs with predictions of adaptive later-life reproductive restraint as a functional explanation for reproductive senescence [20], whereby maternal allocation decreases with age to reduce risks of increased mortality associated with accrued damage due to reproduction [20] or starvation with declining energy dynamics.”

I. 382-384: Alternatively, the lack of reproductive senescence in the wild might be due to a shorter lifespan in the wild.

RESPONSE: Fair point. We have clarified lines 380-382:

“The lack of reproductive senescence in the wild could be linked to shorter lifespans, with wild flies being more susceptible to death from starvation and predation [44,81].”

J. M. Gaillard

Referee: 3

Comments to the Author(s)

In this manuscript the authors describe an optimality model that is used to investigate the conditions that favour hump shaped patterns in maternal investment in offspring throughout the life of the mother.

Overall, I think the model results are interesting and the manuscript is well written.

I therefore have only a few comments and suggestions for the authors. Some of these are just minor details, but changes would still help clarify the m.s.

RESPONSE: We thank the referee for their positive feedback and helpful comments and suggestions which helped us clarify the manuscript.

My only slightly larger/more general concern is about the way you write about results from statistical tests on simulation results. While these results represent the expected population level average maternal investment with respect to age, they do not necessarily represent very well the individual choices or strategies. I have no doubt the authors agree with me on this, but I think it is necessary to clarify this in the text, especially in the discussion.

RESPONSE: This is a good point, and we agree that it is important to consider variation at the individual level. Stable individual differences in state-dependent adaptive behaviour have been shown to occur in another theoretical study in ecological contexts of intermediate favourability [11] which could be similar to what tsetse experience with rich food (vertebrate blood) and high risk (host swatting defences and predation). Our results represent the expected population-level average maternal allocation with respect to age, and it is worth noting that there is no variation within populations in the strategies: for a given parameter set, there is a single optimal strategy, and individual-level variation *in realised behaviour* emerges from stochastic events in the simulations. The individual trajectories are fairly similar as can be seen for example in the Fig.9 of the ESM. There are no strong divergences of behaviour between individuals, and we show the variability in maternal allocation with the error bars.

We now reflect on this more in the discussion at lines 356-364:

“Our results represent the expected population level average maternal allocation with respect to age, which may not necessarily capture the individual strategies. Stable individual differences in state-dependent adaptive behaviour have been shown to occur in another theoretical study in ecological contexts of intermediate favourability [79] which could be similar to what tsetse experience with rich food (vertebrate blood) and high risk (host swatting defences and predation). Although there is no variation within populations in the strategies in our model, for a given parameter set there is a single optimal strategy, individual-level variation in realised behaviour emerges from stochastic events in the simulations. However, there were no strong divergences of behaviour between individuals with individual trajectories being fairly similar (see ESM Fig.9).”

I. 91-92: do you actually investigate stochasticity in energetic costs and feeding success? The way I read your Table 2, it seems that there is no systematic investigation of differences in stochasticity? Is this specified somewhere else?

RESPONSE: We did introduce stochasticity but did not systematically investigate differences in stochasticity and we have now clarified at lines 92-95:

“We introduce stochasticity in energetic costs– in terms of the amount of energy required to forage successfully and individual differences in metabolism – and in feeding success. We systematically assess how allocation is influenced by age-dependence in energetic costs, feeding success, energy intake per successful feeding attempt, and environmentally-driven mortality.”

I.103-104 & I. 138: Although this is at least partially explained later in the ms, I think it would be valuable with a brief explanation of why or how there can be stochasticity in costs of flight and blood digestion (these could easily be thought of as relatively constant for a single individual)

RESPONSE: Each feeding attempt and consequently digestion will be quite different each time for each tsetse, because of the uncertainty in finding a host, host defences, predation risk, bloodmeal volume and so forth.

We have now explained this at lines 92-93:

“We introduce stochasticity in energetic costs– in terms of the amount of energy required to forage successfully and individual differences in metabolism – and in feeding success.”

And lines 102-105:

“Tsetse have access to a rich food supply, vertebrate blood [43], but this can be highly uncertain, requiring finding a host and avoiding its defences (e.g. swatting) [37,44], which introduces stochasticity in the flight duration and distance as well as the bloodmeal volume and hence in the energetic costs of flight and blood digestion [37,39,44].”

I. 155: “fitness h”, is it not better to write expected fitness here?

RESPONSE: Yes, we have added ‘expected’ here.

I. 186-187: It was unclear to me at first that this was the starting reserves at emergence, please rewrite to clarify.

RESPONSE: We have merged the sentences and clarified lines 190-192:

“We simulated the life histories of 1000 mothers (ESM, code modified from [53]) following the optimisation strategy and the reserves at the start of adulthood R_1 , the distribution of which was determined using an iterative procedure as described in [54] (ESM).”

I. 267-268: are you not repeating yourself here?

RESPONSE: We have now updated the text lines 270-271:

“Exploring first the baseline case of the model, the optimal allocation decision is dependent on maternal reserves but independent of age (Figure 1.a, grey line).”

I. 342-343 could you link these empirical patterns more strongly to your model? Maybe through specifying which parameters of your model might be involved? It is unclear to me how relevant these empirical results are to your model results.

RESPONSE: We specified which parameters correspond to which traits lines 331-336:

“The evidence for age-dependence in such traits is wide-ranging across systems from a decrease in energetic costs with an improved lactation ability (e.g. seals [25]), an improved energy transfer efficiency (rats [77]), or reduced metabolic requirements post maturation (tsetse [60,61,69]), to an increase in energy intake with an improved mobility post maturation (tsetse [60,61,70]), or a decrease in energy intake later in life because of gut deterioration (Drosophila [78]) or other physiological deteriorations.”

I.389-396. While I do not strongly disagree with this paragraph in any way, I am not quite convinced it is a good ending as the potential implications mentioned here are quite far from the actual model results and new insight on these matters have yet come from the model.

RESPONSE: We agree with the referee, and we have now removed this paragraph and instead conclude lines 387-399 with the main finding of our model (in explaining increase-then-decrease optimal allocation through age-dependent energy dynamics), and a comment on the generalisability of our framework to other iteroparous breeders (see responses above, in particular to Referee 1 points 3 and 6).

ESM: Would it be possible to use the same name for the time axis in all the figures? Is there a difference between larviposition interval and number of reproductive cycles?

RESPONSE: We thank the referee for pointing it out and we have now used the same name for the time axis in all the figures: “Time (number of reproductive cycles)”.

References

1. Bellman RE. 1954 *The Theory of Dynamic Programming*. Santa Monica, CA: RAND Corporation.
2. Mangel M, Clark CW. 1989 *Dynamic Modeling in Behavioral Ecology*. Princeton University Press. (doi:10.1515/9780691206967)
3. Houston AI, McNamara JM. 1999 Models adaptive behaviour approach based state .
4. McNamara J, Houston A. 1980 The application of statistical decision theory to animal behaviour. *J. Theor. Biol.* **85**, 673–690. (doi:10.1016/0022-5193(80)90265-9)
5. McNamara JM, Houston AI, Barta Z, Scheuerlein A, Fromhage L. 2009 Deterioration, death and the evolution of reproductive restraint in late life. *Proceedings. Biol. Sci.* **276**, 4061–6. (doi:10.1098/rspb.2009.0959)
6. Houston AI, Stephens PA, Boyd IL, Harding KC, McNamara JM. 2007 Capital or income breeding? A theoretical model of female reproductive strategies. *Behav. Ecol.* **18**, 241–250. (doi:10.1093/beheco/arl080)
7. Hargrove JW, Muzari MO, English S. 2018 How maternal investment varies with environmental factors and the age and physiological state of wild tsetse *Glossina pallidipes* and *Glossina morsitans morsitans*. *R. Soc. Open Sci.* **5**, 171739. (doi:10.1098/rsos.171739)
8. Emlen JM. 1970 Age Specificity and Ecological Theory. *Ecology* **51**, 588–601. (doi:10.2307/1934039)
9. Berman M, Gaillard JM, Weimerskirch H. 2008 Contrasted patterns of age-specific reproduction in long-lived seabirds. *Proc. R. Soc. B Biol. Sci.* **276**, 375–382. (doi:10.1098/RSPB.2008.0925)
10. Lord JS, Leyland R, Haines LR, Barreaux AMG, Bonsall MB, Torr SJ, English S. 2021 Effects of maternal age and stress on offspring quality in a viviparous fly. *Ecol. Lett.* , ele.13839. (doi:10.1111/ELE.13839)
11. Luttbeg B, Sih A. 2010 Risk, resources and state-dependent adaptive behavioural syndromes. *Philos. Trans. R. Soc. B Biol. Sci.* **365**, 3977–3990. (doi:10.1098/RSTB.2010.0207)

Appendix B

University of
BRISTOL

Dr Antoine Barreaux
Research Associate

School of Biological Sciences, Life Sciences Building
email:antoine.barreaux@gmail.com

18/01/22

Dear Prof Kruuk

We are resubmitting our research manuscript entitled "*Incorporating effects of age on energy dynamics predicts non-linear maternal allocation patterns in iteroparous animals*". We are thankful that our manuscript has been accepted for publication in *Proceedings of the Royal Society B*.

We are grateful to the associate editor and reviewers for the minor revisions suggested and their recommendation for publication. We have made the suggested changes in light of each reviewer's comments. Our detailed responses are provided below.

We describe original work that is not being considered for publication in any other journal. We declare having no competing interests. All the authors gave final approval for publication and submission to *Proceedings B* of the final manuscript.

We appreciate your time and are looking forward to your response.

Best regards,

Antoine Barreaux, on behalf of all authors

Associate Editor

Board Member: 1

Comments to Author:

The authors have done an excellent job in responding to reviewers' comments. I do encourage the authors to address the remaining issues raised by the reviewers. In particular, J.M. Gaillard explains in detail why he is concerned about the use of quadratic models in this case. I think the authors will have little problem in addressing these points and I think this paper will advance the field.

RESPONSE: We thank the associate editor for their time, their interest in our research and their support throughout the submission process. We have made sure to answer all the remaining helpful queries made by the two referees. Please find our detailed answer below.

Referee: 1

Comments to Author:

Dear Editor,

I reviewed a previous version of this manuscript and the authors have done an excellent job responding to my previous comments of the other reviewers. Nevertheless, I have a few comments regarding the revised manuscript. The first two issues are relatively minor. The third is more substantial.

RESPONSE: We thank the referee for their interest in our manuscript, their appreciation of our previous answers, and the last few comments on our approach and conclusions.

1. Line 22, line 388 (and elsewhere): There are two major explanations for the evolution of senescence: optimality and mutation pressure (e.g., Partridge and Barton 1993). The optimality explanation posits that ageing could evolve as part of an optimal life history in which there is an antagonism between performance early in life and performance late in life. In contrast, mutation pressure could lead to ageing

because selection against late-acting deleterious mutations will be weak. These two explanations cannot be distinguished based upon the relationship between parental allocation and parental age. Thus, I do not think it is correct to say that “optimal” allocation is hump shaped across ages in diverse taxa. The hump-shaped pattern may be widespread, but in most cases we do not know whether the shape of this curve can be explained by the optimality hypothesis or the mutation pressure hypothesis.

RESPONSE:

This is a fair point, in terms of distinguishing the hump-shaped pattern across empirical studies.

We have clarified lines 24 and 425 by removing the word “optimal”. That said, our model is focused on the optimality explanation, as we assume no mutations and as suggested by referee 2, we have clarified that assumption lines 246-248:

“As such, we do not impose a decline in selection in later life as all offspring are potentially equally valuable at all ages (for for the same maternal allocation), and we assume there are no mutations.”

And we have also clarified the start of the discussion lines 338-343:

“Our model predicts optimal maternal allocation of resources is non-linear with age, when there is age-dependence in key drivers of energy dynamics. Such a non-linear relationship between parental allocation and age has been found in many species. Our model assumes no mutations and hence provides further theoretical insight into the drivers of age-dependent allocation in terms of optimal life-history allocation, although we acknowledge that similar patterns can also arise from changes in mutation pressure which are not considered in our model.”

2. Line 51: genes should be alleles.

RESPONSE: Corrected.

3. Line 80-87 and 236-238: I have a hard time reconciling these two sections. In the first passage, the authors acknowledge that offspring quality may decline with parental age, and as a consequence later-born offspring might have lower reproductive value to parents than earlier-born offspring. The authors suggest that this pattern is not accounted for in current models of senescence, which focus on fecundity and not offspring quality. However, in the second passage the authors state that they are assuming that the reproductive value of earlier-born and later-born offspring is the same. I understand that the goal is to examine whether maternal allocation strategies can generate hump-shaped allocation curves without invoking a decline in the strength of selection with age. However, it is not clear whether / how the model predictions will change if the assumption that all offspring contribute equally to parental fitness is relaxed.

RESPONSE:

We appreciate that the phrasing of these two sections may have caused confusion. To clarify, we assume here that the reproductive value of earlier-born and later-born offspring is *potentially* the same for a given level of maternal allocation as we do not want to force offspring quality to decrease with age by invoking a decline in the strength of selection with age. However, mothers may vary allocation of resources to offspring with age, which will then result in offspring of different quality and may lead to reproductive senescence if offspring quality decreases with maternal age. So, offspring quality may indeed vary with age depending on the optimal strategy.

We agree that it could be interesting to relax the assumption that all offspring potentially contribute equally to parental fitness whatever the parental age, yet this is beyond the scope of our current study. Making assumptions about the offspring as well as parents would make the model considerably more complex and would require careful thought about how these would impact model predictions. We hope our model will inspire such future work exploring the effects of varying assumptions about the offspring.

We clarified lines 246-250 by adding that all offspring are potentially equally valuable, but that offspring quality may change if maternal allocation changes with age:

“As such, we do not impose a decline in selection in later life as all offspring are potentially equally valuable at all ages (for for the same maternal allocation), and we assume there are no mutations. However, mothers may vary allocation of resources to offspring with age, which will then result in

offspring of different quality and may lead to reproductive senescence if offspring quality decreases with maternal age.”

We added lines 371-374 in the discussion:

“Imposing a declining selection with age by relaxing the hypothesis that all offspring are equal may potentially nuance our predictions about non-linear parental allocation. We hope our model will inspire future work on age-dependent allocation under varying assumptions about offspring quality.”

Referee: 2

Comments to the Author(s)

I warmly thank the authors for having provided both detailed responses to comments and a carefully-revised manuscript. I was really convinced by the changes performed (in particular the link between the authors' model and the capital-income breeder continuum). However, I still have two concerns about this paper, which should be easy to solve.

RESPONSE: Thank you for warm thanks and positive and helpful comments and suggestions.

First, the two assumptions the authors clearly stated in their responses to the first reviewer's comments should be presented and discussed more thoroughly in the paper itself. The « as » in l.322 in the version with tracked changes should be replaced with « as we assume ». Indeed, there appear to be quite strong assumptions because in the real world (1) the timing of reproduction throughout the lifetime matters a lot in terms of fitness but in case the population is stationary, and (2) mutations should occur soon or later.

RESPONSE: We appreciate these are key assumptions and we have clarified these, and their implications, in the revised version. As specified above in our answer to referee 1 we have added more details about our assumptions about the absence of mutations and we emphasize that the value of all offspring is *potentially* equal across parental age.

We have added that we assume there are no mutations in our model lines 246-248 and we have clarified the start of the discussion lines 338-343 how important it is when coming back to the general theory of ageing and the distinction between optimality and mutation pressure explanations. We clarified lines 246-250 that all offspring are potentially equally valuable, but that offspring quality may change if maternal allocation changes with age. We also discussed relaxing that assumption lines 371-374.

Second, I still disagree with the use of quadratic models to infer an initial increase of reproductive performance until a peak or plateau followed by a decrease of reproductive performance. To illustrate my concern, consider the following age-specific trajectory of performance (rp):

```
age<-c(1,2,3,4,5,6,7,8,9,10,11)
```

```
rp<-c(0.01,0.05,0.20,0.60,1.2,1.5,1.6,1.7,1.75,1.75,1.75)
```

This rp trajectory is characterized by a continuous increase of reproduction with age until a plateau is reached. There is no evidence of any reproductive senescence from this trajectory.

Now fit a quadratic model:

```
mod<-lm(rp ~ age + I(age^2))
```

and look at the estimates:

Coefficients:

	Estimate	Std. Error	t value	Pr(> t)
(Intercept)	-0.708182	0.206638	-3.427	0.008992 **
age	0.469056	0.079144	5.927	0.000351 ***
I(age^2)	-0.021853	0.006424	-3.402	0.009333 **

The outcome is a statistically significant negative quadratic coefficient in absence of reproductive senescence. This clearly demonstrates that a negative quadratic term cannot be interpreted in terms of decreasing performance in late life. Although additional models ran by the authors convinced me that

their findings are robust, I would ask the authors to add an explicit statement about the caution required when interpreting quadratic parameters in terms of senescence to avoid the pitfall mentioned just above.

J.M. Gaillard

RESPONSE: Thank you for providing this effective example. We agree that caution is required when interpreting quadratic parameters in terms of senescence to infer an initial increase of reproductive performance until a peak or plateau followed by a decrease of reproductive performance.

We added the following statement in the methods lines 266-273:

“It is worth noting that caution is required when interpreting quadratic parameters in terms of senescence to infer an initial increase of reproductive performance until a peak or plateau followed by a decrease of reproductive performance. This is because the presence of a statistically significant negative quadratic coefficient does not necessarily indicate a hump-shaped curve but can also represent a case of diminishing returns where allocation plateaus in later-life but does not decline (hence, no reproductive senescence).”